# Serum/glucocorticoid-inducible kinase 1 deficiency induces NLRP3 inflammasome activation and autoinflammation of macrophages in a murine endolymphatic hydrops model

Dao-Gong Zhang [1,2,3,4], Wen-Qian Yu[1,3,4], Jia-Hui Liu [1,3,4], Li-Gang Kong [1,2,3], Na Zhang [1,3], Yong-Dong Song [1,2,3], Xiao-Fei Li [1,2,3], Zhao-Min Fan[1,2,3], Ya-Feng Lyu [1,2,3], Na Li [1,3] ✉ & Hai-Bo Wang [1,2,3] ✉

Ménière's disease, a multifactorial disorder of the inner ear, is characterized by severe vertigo episodes and hearing loss. Although the role of immune responses in Ménière's disease has been proposed, the precise mechanisms remain undefined. Here, we show that downregulation of serum/glucocorticoid-inducible kinase 1 is associated with activation of NLRP3 inflammasome in vestibular-resident macrophage-like cells from Ménière's disease patients. Serum/glucocorticoid-inducible kinase 1 depletion markedly enhances IL-1β production which leads to the damage of inner ear hair cells and vestibular nerve. Mechanistically, serum/glucocorticoid-inducible kinase 1 binds to the PYD domain of NLRP3 and phosphorylates it at Serine 5, thereby interfering inflammasome assembly. $Sgk^{-/-}$ mice show aggravated audiovestibular symptoms and enhanced inflammasome activation in lipopolysaccharide-induced endolymphatic hydrops model, which is ameliorated by blocking NLRP3. Pharmacological inhibition of serum/glucocorticoid-inducible kinase 1 increases the disease severity in vivo. Our studies demonstrate that serum/glucocorticoid-inducible kinase 1 functions as a physiologic inhibitor of NLRP3 inflammasome activation and maintains inner ear immune homeostasis, reciprocally participating in models of Ménière's disease pathogenesis.

Ménière's disease (MD), described in 1861, is a complex heterogeneous syndrome characterized by episodic vertigo, along with fluctuating low-to-medium frequencies of sensorineural hearing loss (SNHL), tinnitus, or aural fullness, and may affect one or both ears (bilateral MD)[1].

The pathophysiology of MD is associated with endolymphatic hydrops (EH) in the cochlear duct and vestibular end organs (VEO)[2,3], although EH is believed to be an epiphenomenon or byproduct of the underlying disease rather than a causal factor[4]. Numerous underlying factors

[1]Department of Otolaryngology-Head and Neck Surgery, Shandong Provincial ENT Hospital, Shandong University, Jinan, Shandong, China. [2]Shandong Provincial Vertigo & Dizziness Medical Center, Jinan, Shandong, China. [3]Laboratory of Vertigo Disease, Shandong Institute of Otorhinolaryngology, Shandong Second Provincial General Hospital, Jinan, Shandong, China. [4]These authors contributed equally: Dao-Gong Zhang, Wen-Qian Yu, Jia-Hui Liu. ✉e-mail: linda831223@163.com; whboto11@163.com

interact in MD, including anatomical variations in the temporal bone, genetics, autoimmunity, migraine, altered intralabyrinthine fluid dynamics, and cellular and molecular mechanisms[5]. It is estimated that a third of patients with MD have immune dysfunction, however, the immunological mechanisms underlying MD remain poorly understood[6].

Previously, the inner ear was believed to be devoid of immune responses as it is separated by a blood/labyrinth barrier. However, recently, the cochlea and VEO were shown to contain a large population of macrophages, the functions of which in inner ear disease remain undefined[7–10]. Tissue-resident macrophages play critical roles in organ-specific immune regulation. Accumulating evidence suggests that proinflammatory cytokines secreted by macrophages may predispose the inner ear microenvironment to immune maladaptation[10,11]. Inner ear macrophages can be recruited from blood-borne monocytes to hair cells following damage induced by noise, ototoxic drugs, aging, and diphtheria toxin-induced selective hair cell degeneration[8]. These macrophages may release interferons, inflammatory cytokines, and chemokines via the complement cascade. However, whether vestibular-resident macrophage-like (VRML) cells mediate local innate immune responses in hearing loss disorders, such as sudden SNHL and MD, remains to be elucidated.

A case-control study reported that the levels of proinflammatory cytokines such as interleukin (IL)–1β, IL-1RA, tumor necrosis factor alpha (TNF-α), and IL-6 were elevated in the supernatant of peripheral blood mononuclear cells obtained from 21% of MD patients[12]. Additionally, polymorphisms in *IL1A* and *IL-1R1* are associated with MD[13,14]. IL-1β mediates autoinflammation, which may involve the NLR family pyrin domain containing 3 (NLRP3) inflammasome. NLRP3 complexes with apoptosis-associated speck-like protein containing a CARD (ASC) to form a caspase-1-activating complex that cleaves and activates IL-1β[15]. Dysregulation of the NLRP3 inflammasome has been linked with several chronic inflammatory, infectious, and autoimmune diseases[16], including Alzheimer's disease[17], human immunodeficiency virus (HIV) infection[18], and atherosclerosis[19]. In fact, the NLRP3 inflammasome can be activated in macrophage or monocyte-like cells residing in the mouse cochlea, resulting in IL-1β secretion[9].

Intratympanic steroid injections are a non-ablative, safe, and effective treatment for patients with refractory MD[20]. Steroids affect ion homeostasis and induce immune modulation via glucocorticoid and mineralocorticoid receptors[21]. Both types of receptors are present in the vestibular and cochlear systems. Glucocorticoids exert their effects partly via serum/glucocorticoid-inducible kinase 1 (SGK1), whose expression was shown to be upregulated in rat mammary tumor cells following serum and glucocorticoid treatment[22]. SGK1 belongs to the serine/threonine kinase family and is ubiquitously expressed in various tissues, including the inner ear[23,24]. Importantly, SGK1 expression can be regulated at different levels by growth factors, glucocorticoids, and inflammatory factors, and it can regulate multiple ion channels, membrane transporters, cellular enzymes, and transcription factors involved in various physiological processes[25]. Although the SGK1 level is regulated in the cochlea in response to glucocorticoids[26], its role in MD is not entirely understood.

Animal models of EH have provided a basic understanding of MD pathogenesis. Models of chronic EH have been developed using surgical ablation of the endolymphatic sac or systemic medication such as aldosterone, vasopressin, or lipopolysaccharide (LPS) for conducting vestibular research[27]. For example, LPS induces moderate to severe hearing loss and EH in the cochlear duct, with severe cellular infiltration[28]. However, mouse models have not provided sufficient insights to completely understand immune pathogenesis of MD phenotypes.

In this study, we show that SGK1, which is reduced in VRML cells from MD patients, is a negative regulator of NLRP3 inflammasome activation. SGK1 depletion markedly enhances IL-1β production which

leads to the damage of inner ear hair cells and vestibular nerve. Further studies demonstrate that SGK1 protects macrophages from pyroptosis by interacting with NLRP3 and phosphorylating its Serine 5 (S5) to inhibit inflammasome assembly. In addition, we show that SGK1 deficiency causes an exacerbation of EH and audiovestibular symptoms, with exaggerated NLRP3 activation and proinflammatory cytokine production in LPS-induced EH model, which is ameliorated by MCC950, an inhibitor of NLRP3. Together, our findings uncover a mechanism by which SGK1 controls NLRP3 inflammasome activation link to macrophage autoinflammation in a murine EH model and suggest a potential therapeutic for Ménière's disease.

## Results

### Decrease in SGK1 expression is associated with inflammasome activation in VRML cells

To investigate the relationship between inflammation and MD pathology, we evaluated the number of VRML cells in clinical tissue samples of VEO from patients with MD and acoustic neuroma (AN). As shown in Fig. 1a, the number of IBA1-positive macrophages in different regions (ampulla and macula) of the VEO was significantly higher in the MD group than in the AN group. Next, we analyzed the local expression of the downstream NLRP3 inflammasome activation markers including IL-1β, IL-18 and caspase-1 in clinical tissue samples. Immunohistochemistry of vestibular tissue sections confirmed that IL-1β, IL-18, and caspase-1 expression increased in the ampulla and macula from patients with MD (Fig. 1b, Supplementary Fig. 1a). Analysis of mRNA levels revealed a significantly increased transcription of inflammasome-associated markers in the VEO in the MD group compared to that in the AN group (Fig. 1c). IL-1β, IL-18, and caspase-1 levels were elevated systemically in the plasma of MD patients during the active phase compared with those in the plasma of age matched controls (Fig. 1d). Furthermore, we recruited 20 patients during the quiescent phase and found that IL-1β levels were elevated in 25% of them (Supplementary Fig. 1b).

To determine whether SGK1 was involved in MD, we evaluated SGK1 expression in the VEO of patients with MD and AN. Primary VRML cells were isolated from patients with MD and AN (Supplementary Fig. 1c). Analysis of mRNA levels revealed significant downregulation of *SGK1* expression in the VEO or primary VRML cells from patients with MD compared to that in the AN group (Fig. 1e, f). These findings were further confirmed using confocal microscopy (Supplementary Fig. 1d). Western blot analysis confirmed that the protein levels of SGK1 in primary VRML cells from the MD group were lower than those in the AN group, while the expression of NLRP3 was higher than that in the AN group (Fig. 1g). Staining VEO samples of patients for IL-1β, SGK1, and the macrophage marker IBA-1 revealed a high expression of IL-1β with a decreased expression of SGK1 in VRML cells of the MD group as compared to that in the AN group (Fig. 1h). These analyses indicate that SGK1 downregulation is associated with NLRP3 inflammasome activation in the VEO and that it may participate in the immune pathophysiology of MD. Activation of the NF-κB pathway has been shown to downregulate SGK1 expression[29]; herein, the expression of NF-κB p65 was significantly upregulated in the VEO or primary VRML cells from patients with MD compared that in patients with AN (Supplementary Fig. 1e, f). These results suggest that downregulation of SGK1 in MD may be attributed to NF-κB pathway activation.

Hearing loss is frequently caused by defects in hair cells in the basilar membrane[30], and the maintenance of balance depends on the normal structure and function of the vestibular system[31]. The level of IL-1β, the major effector molecule of the NLRP3 inflammasome, was significantly elevated in the VEO of patients with MD. Moreover, we have confirmed that CD68-positive macrophages were widely distributed in all turns of the cochlea, primarily in the cochlear modiolus, stria vascularis, spiral ligament, spiral ganglion, and basilar membrane (Supplementary Fig. 1g). We next investigated whether IL-1β produced

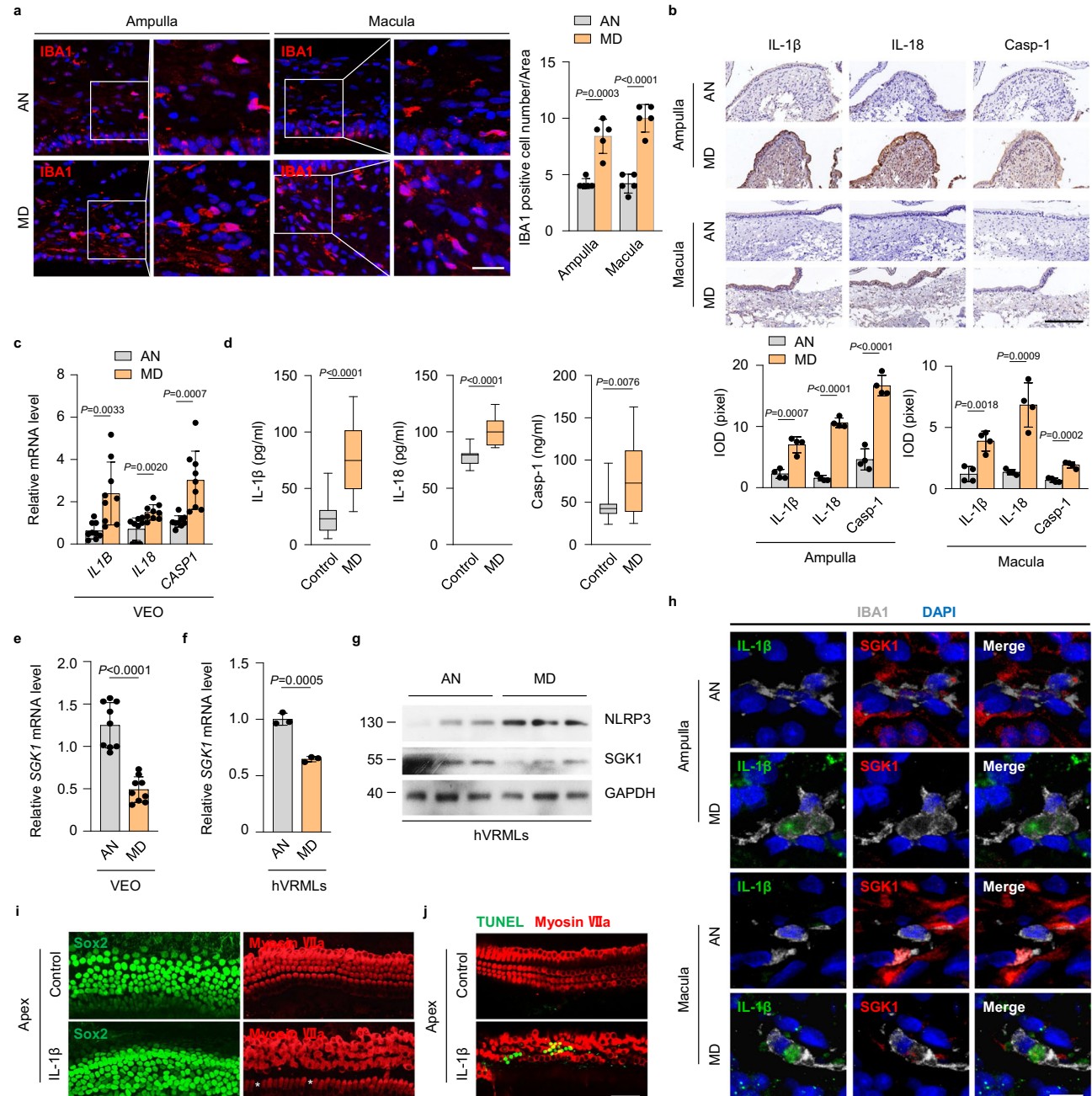

**Fig. 1 | Decrease in SGK1 expression is associated with inflammasome activation in VRML cells from MD patients. a** Representative confocal microscopy images showing IBA1 (red) and DAPI (blue) staining in the ampulla and macula of AN (*n* = 5) and MD patients (*n* = 5). Analysis of images is shown on the right, quantifying the number of IBA1-positive cells per area. The right panels represent 2-fold enlargements of the white-boxed areas in the left panels. Scale bar = 25 μm. **b** Representative images of immunohistochemical staining and densitometric quantification of IL-1β, IL-18, and Casp-1 in the ampulla and macula of AN (*n* = 4) and MD patients (*n* = 4), scale bar = 100 μm. **c** Quantitative real-time PCR analysis of *IL1B*, *IL18*, and *CASP1* mRNA in VEO of AN (*n* = 9) and MD patients (*n* = 9). **d** ELISA of plasma IL-1β, IL-18, and Casp-1 levels in controls (*n* = 20) and MD patients (*n* = 20). **e** Quantitative real-time PCR analysis of *SGK1* mRNA in VEO of AN (*n* = 9) and MD patients (*n* = 9). **f** Quantitative real-time PCR analysis of *SGK1* mRNA in VRML cells of AN and MD patients (*n* = 3). **g** Representative images of western blotting detecting NLRP3 and SGK1 in VRML cells of AN and MD patients (*n* = 3). **h** Representative confocal microscopy images showing IL-1β (green), SGK1 (red), IBA1 (gray), and DAPI (blue) staining in the ampulla and macula of AN and MD patients, scale bar = 10 μm. **i** Immunofluorescence staining for Sox2 (green) and Myosin VIIa (red) in basilar membrane of neonatal mice in vitro, after treatment with IL-1β (10 ng/mL, 24 h). White asterisks indicate inner hair cell loss. Scale bar = 50 μm. **j** Immunofluorescence staining for TUNEL (green) and Myosin VIIa (red) in the basilar membrane at P3, after treatment with IL-1β (10 ng/mL, 24 h), scale bar = 50 μm. In each box plot, the central mark indicates the median, the bottom and top edge of the box indicate the interquartile range, and the whiskers represent the max and minimum data point. Results are presented as mean ± SD. Statistical analyses were carried out via two-sided *t*-test for **a–f**. VEO vestibular end organs, hVRMLs human vestibular-resident macrophage-like cells, Casp-1 caspase-1. Source data are provided as a Source Data file.

by macrophages affected the inner ear hair cells and vestibular neurons. We exposed the basilar membrane of all turns from the mouse cochlea to IL-1β in vitro. Compared to the control group, cochlear hair cells (Myosin VIIa positive) exposed to IL-1β showed disordered arrangement, cell loss, and increased apoptosis in the apical turn (Fig. 1i, j), while those in the middle and basal turns showed no obvious changes (Supplementary Fig. 1h). No significant changes in the cochlear supporting cells (Sox2 positive) were detected between the

control and IL-1β groups (Fig. 1i and Supplementary Fig. 1h). The expression of cleaved caspase-3 dramatically increased in IL-1β-treated ampulla and macula, similar to that in the vestibular ganglion (Supplementary Fig. 1i). Together, IL-1β can damage cochlear and vestibular hair cells, as well as the vestibular nerves, leading to auditory and vestibular dysfunction.

## SGK1 limits NLRP3 inflammasome activation and protects from pyroptosis in macrophages

Resident macrophage/monocyte-like cells in the mouse cochlea express NLRP3, can secrete IL-1β, and may thus mediate local autoinflammation by activating the NLRP3 inflammasome[9]. To investigate the potential role of SGK1 in regulating NLRP3 inflammasome signaling, we isolated mouse VRML (mVRML) cells from sgk1 knocked out (sgk1[−/−]) mice and littermate sgk1[+/+] mice (Supplementary Fig. 2a). We observed that the upregulation of endogenous NLRP3 expression in mVRML cells was concentration-dependent and time-dependent, and the effect was optimal at 200 ng/ml and 3 h (Supplementary Fig. 2b). Quantitative PCR (qPCR) revealed that sgk1[−/−] mVRML cells stimulated with LPS minus or plus ATP showed significant elevation of Nlrp3, Casp1, and Il1b mRNA expression compared to those in the controls (Fig. 2a). Augmented protein levels of NLRP3, pro caspase-1, caspase-1 (p20), pro IL-1β, and cleaved IL-1β in sgk1[−/−] mVRML cells were also confirmed using western blotting (Fig. 2b). ELISA demonstrated a significant increase in secreted caspase-1 and IL-1β production in sgk1[−/−] mVRML cells as compared with that in sgk1[+/+] controls under inflammasome-activating conditions (Fig. 2c), indicating that SGK1 suppresses caspase-1 (p20) and IL-1β cleavage secretion. Immunofluorescence imaging showed the increase in NLRP3 expression in mVRML cells of sgk1[−/−] mice than sgk1[+/+] mice (Fig. 2d). Of note, NLRP3 was localized not only in the nucleus but also in the cytoplasm (Fig. 2d and Supplementary Fig. 2c).

To avoid cellular heterogeneity of mVRML cells, we knocked down SGK1 in THP-1 macrophages. Consistent with the results obtained using sgk1[−/−] mVRML cells, the mRNA expression levels of NLRP3, CASP1 and IL1B increased in SGK1 small interfering RNA (siRNA) knockdown THP-1 (Fig. 2e). As shown in Fig. 2f, THP-1 cells treated with LPS minus or plus ATP showed higher NLRP3 expression and higher levels of cleavage caspase-1 and IL-1β in SGK1-depleted cells than control cells. Furthermore, the caspase-1 and IL-1β secretion was significantly higher in SGK1-depleted THP-1 cells than in control cells when exposed to inflammatory threats (Fig. 2g, Supplementary Fig. 2d). Immunofluorescence imaging revealed increase in ASC (Fig. 2h) and NLRP3 (Supplementary Fig. 2e) expression and its assembly into specks in SGK1 depleted THP-1 cells stimulated with LPS and ATP. To further confirm this finding, we also expressed GFP-NLRP3 and FLAG-SGK1 in HEK-293T cells and found that overexpressed SGK1 perturbed the assembly of NLRP3 into specks after stimulation with LPS and ATP (Supplementary Fig. 2f). Taken together, the data suggest that SGK1 suppresses the cleavage of caspase-1 and IL-1β presumably by suppressing the NLRP3 inflammasome.

Pyroptosis involves caspase-1-mediated monocyte death (via pyroptosomes)[32]. Propidium iodide (PI), a nuclear and chromosome counterstain, is not permeant to live cells; hence, it is commonly used to detect pyroptotic cells[33]. Annexin V and PI double staining was performed to assess the role of SGK1 in pyroptosis. The results showed increased apoptotic and pyroptotic populations in SGK1-depleted cells compared to those in the control when exposed to inflammatory threats; 10.30% of cells exhibited pyroptotic signatures, showing Annexin V(+)/PI(+) status, while 9.78% of cells underwent apoptosis, showing Annexin V(+)/PI(−) status (Fig. 2i, Supplementary Fig. 2g). These results suggest that SGK1 protects cells from inflammation-induced cell death. Next, we blocked caspase activity using the pan-caspase inhibitor, z-VAD-FMK, or the selective caspase-1 inhibitor, Ac-YVAD-cmk, to determine whether the inflammatory cell death observed in SGK1-depleted cells was caspase-dependent. Results

showed that z-VAD-FMK and Ac-YVAD-cmk blocked pyroptosis (7.7 and 9.19%, respectively) and reduced apoptosis (5.76 and 6.91%, respectively) induced by SGK1 depletion (Fig. 2i), suggesting that SGK1 suppressed caspase-1 dependent inflammatory cell death. Recent studies have shown that gasdermin D (GSDMD), the substrate for caspase-1 and caspases 4, 5, and 11, is the effector protein of pyrolysis[34]. In agreement with the caspase-1 cleavage data presented in Fig. 2b, f, GSDMD cleavage was higher in SGK1-depleted cells than in control cells, while GSDMD cleavage decreased when z-VAD-FMK or Ac-YVAD-cmk was used (Fig. 2j). The DNA repair enzyme, poly ADP-ribose polymerase 1 (PARP1), is a substrate for caspases. PARP1 cleavage increased in SGK1-depleted cells and was inhibited when z-VAD-FMK or Ac-YVAD-cmk was used (Fig. 2j). These results suggest that SGK1 protects cells from inflammation-induced cell death.

## SGK1 co-localizes and interacts with NLRP3

To investigate the mechanism via which SGK1 inhibits NLRP3 inflammasome activation, we examined the subcellular co-localization of SGK1 with NLRP3. In immunofluorescence assays, endogenously and exogenously expressed SGK1 co-localized with NLRP3 in LPS-primed THP-1 (Fig. 3a) and HEK-293T cells (Fig. 3b). Similarly, in LPS-primed mVRML cells (Fig. 3c) and vestibular tissues from MD patients (Fig. 3d), SGK1 co-localized with NLRP3. To confirm the physical association between SGK1 and NLRP3, we incubated total cell lysates of LPS-stimulated THP-1 cells with a specific anti-SGK1 antibody. The co-immunoprecipitated complex could be immunoblotted with antibodies against NLRP3, which showed that SGK1 interacted with NLRP3 (Fig. 3e). Immunoprecipitation with antibodies against NLRP3, followed by immunoblotting with antibodies against SGK1, confirmed these interactions (Fig. 3e). Similarly, exogenous experiments revealed that SGK1 interacted with NLRP3 (Fig. 3f). The GST pull-down assays showed that the purified SGK1 directly interacted with NLRP3, confirming the results of the co-immunoprecipitation (Co-IP) assays (Fig. 3g). To analyze whether the interaction between SGK1 and NLRP3 depended on LPS priming, we assessed interactions between SGK1 and NLRP3 in resting, LPS-primed, and ATP-stimulated THP-1 cells. Endogenous SGK1 and NLRP3 co-immunoprecipitated under all conditions; however, the binding of SGK1 to NLRP3 considerably reduced in the presence of LPS (Fig. 3h). These results indicate that SGK1 acts as a physiological "gatekeeper" of NLRP3 inflammasome activation under normal conditions. Taken together, these data suggeste that SGK1 co-localizes and interacts with NLRP3 directly.

To further understand the details of the NLRP3-SGK1 interaction, we performed a domain-mapping experiment. NLRP3 is composed of PYRIN (PYD), NACHT (nucleotide binding domain or NBD), and LRR domains (Fig. 3i). We transfected HEK-293T cells with plasmids expressing a full-length SGK1 and one of various NLRP3 domains. After 48 h, the cells were used for immunoblotting analysis. SGK1 interacted predominantly with PYD, but negligibly with the LRR or NACHT domains (Fig. 3i). Collectively, these data indicate that SGK1 interacts with NLRP3 by binding to the PYD domain.

## SGK1 phosphorylates NLRP3 at serine 5 and inhibits inflammasome assembly

SGK1 belongs to the AGC serine/threonine kinase family, highly homologous to AKT that mediates phosphorylation of several proteins[35]. Earlier reports have demonstrated that AKT is able to directly phosphorylate NLRP3 at residues S5, limiting NLRP3 oligomerization[36]. Our Co-IP results showed that SGK1 interacted with NLRP3 by binding to the PYD domain. Therefore, we hypothesized that SGK1 plays a critical role in regulating NLRP3 inflammasome activation by directly phosphorylating NLRP3 at S5. To test this hypothesis, we isolated VRML cells from sgk1[+/+] and sgk1[−/−] mice. After LPS priming and ATP stimulation, we prepared cell lysates for immunoprecipitating endogenous NLRP3; immunoblot analysis of p-Ser showed diminished

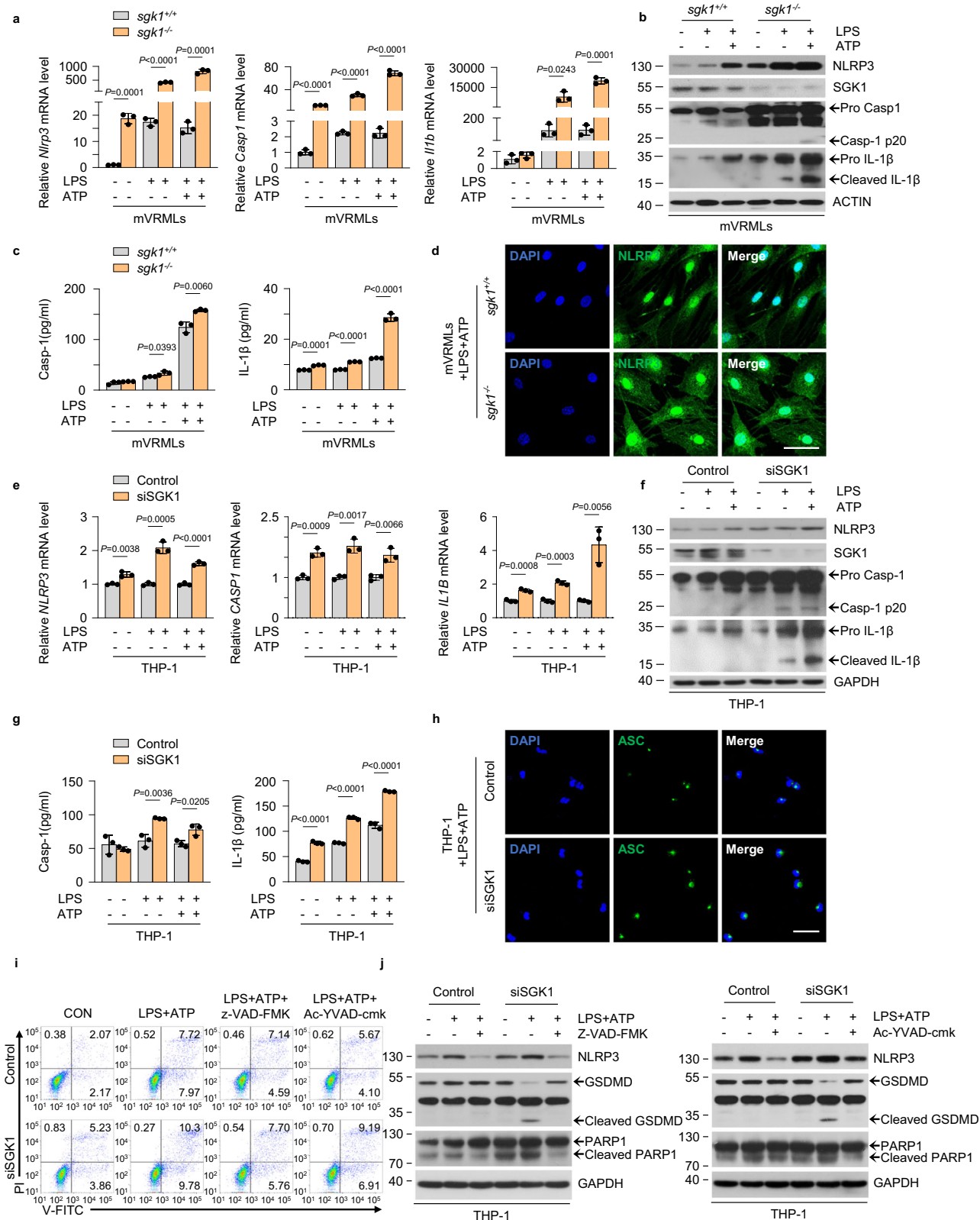

phosphorylation of NLRP3 in *sgk1⁻/⁻* VRML cells compared with that in *sgk1⁺/⁺* VRML cells (Fig. 4a). Similarly, this reduction in NLRP3 serine phosphorylation was observed in SGK1-knockdown THP-1 cells (Fig. 4b). We then established an in vitro kinase assay using recombinant inactive SGK1 (S422A) and active SGK1 (S422D) and recombinant GST-NLRP3 as a substrate. Immunoblot analysis demonstrated that serine phosphorylation of NLRP3 was induced by the active SGK1

protein but not by inactive SGK1 (Fig. 4c). These data show that SGK1 indeed phosphorylates NLRP3 in an exogenous system and in endogenous ATP-induced conditions.

NLRP3 contains multiple serine residues, including S5, S161, S163, S198, S295, S334, S387, S436, and S728[37]. Previous domain segmentation experiments have shown that SGK1 can bind to the PYD region of NLRP3, which contains the S5 residues. To test whether SGK1 directly

**Fig. 2 | SGK1 limits NLRP3 inflammasome activation and protects from pyroptosis. a–d** Primary mVRML cells from *sgk1^{+/+}* or *sgk1^{−/−}* mice upon stimulation with LPS (200 ng/mL, 3 h) alone or with ATP (5 mM, 30 min). **a** Quantitative real-time PCR of indicated genes (*n* = 3 biologically independent experiments). **b** Cell lysates were subjected to western blotting. **c** ELISA showing Casp-1 and IL-1β secretion (*n* = 3 biologically independent experiments). **d** Confocal microscopy of NLRP3 (green) and DAPI (blue) staining, scale bar = 50 μm. **e–h** Control and *SGK1* siRNA-transfected THP-1 cells were stimulated with LPS (200 ng/mL, 3 h) alone or with ATP (5 mM, 30 min). **e** Quantitative real-time PCR of indicated genes (*n* = 3 biologically independent experiments). **f** Cell lysates were subjected to western

blotting. **g** ELISA showing Casp-1 and IL-1β secretion (*n* = 3 biologically independent experiments). **h** Confocal microscopy of ASC (green) and DAPI (blue) staining, scale bar = 50 μm. **i** Dot plot showing flow cytometry analysis of control and *SGK1* siRNA-transfected THP-1 cells with or without LPS and ATP in the presence or absence of z-VAD-FMK (20 μM, 3.5 h) or Ac-YVAD-cmk (40 μM, 3.5 h). **j** Western blotting with control and *SGK1* siRNA-transfected THP-1 cells with or without LPS and ATP in the presence or absence of z-VAD-FMK (20 μM, 3.5 h) or Ac-YVAD-cmk (40 μM, 3.5 h). Results are presented as mean ± SD. Statistical analyses were carried out via two-sided *t*-test for **a**, **c**, **e**, **g**. mVRMLs, mouse vestibular-resident macrophage-like cells; Casp-1 caspase-1. Source data are provided as a Source Data file.

phosphorylates NLRP3 at S5, immunoprecipitated active SGK1 was mixed with recombinant NLRP3 or its non-phosphorylated mutant NLRP3 S5A. Serine phosphorylation of the NLRP3 S5 mutant showed a significant reduction compared with wild-type NLRP3 (Fig. 4d), suggesting that SGK1 directly phosphorylates NLRP3 on S5.

Latz et al. found that PYD phosphorylation regulates charge-charge interactions between two PYDs essential for NLRP3 activation[38]. Therefore, we investigated whether SGK1 affected NLRP3 oligomerization. As shown in Fig. 4e, NLRP3 oligomerization increased significantly after stimulation with LPS and ATP; SGK1 knockdown further increased NLRP3 oligomerization after treatment with LPS minus or plus ATP. Thus, NLRP3 activation possibly forms a molecular seed that nucleates the formation of helical fibrils of ASC via interacting with PYD. Previous studies have shown that introduction of negative charges via S5 phosphorylation disturbs the interaction between NLRP3 and ASC[38,39]. To further understand the effect of SGK1 on NLRP3-ASC interactions, we performed Co-IP assays and found that the interaction between NLRP3 and ASC reduced when SGK1 was overexpressed in HEK-293T cells (Fig. 4f). These interactions were significantly enhanced when SGK1 was knocked down (Fig. 4g). Taken together, these data show that SGK1 restricts NLRP3 oligomerization and reduces the interaction between NLRP3 and ASC.

### SGK1 deficiency exacerbates LPS-induced EH and audio-vestibular symptoms in vivo

The LPS-induced EH model[28] presents an opportunity to determine how the effects of SGK1 on the NLRP3 inflammasome affect a clinically relevant MD process. We established a LPS-induced EH mouse model from which the cochleae were collected. Immunofluorescence staining showed NLRP3-positive cells scattered in all turns, and more NLRP3-positive staining was detected in the apex than in the base (Supplementary Fig. 3a). Furthermore, we postauricularly (p.a.) injected LPS to age- and sex-matched *sgk1^{−/−}* and littermate *sgk1^{+/+}* mice for 3 consecutive days, and then assessed the features of EH and audiovestibular symptoms induced (Supplementary Fig. 3b, c). After administration of LPS, *sgk1^{−/−}* mice showed more severe EH in cochlear half-turns II–IV than did *sgk1^{+/+}* mice, but not in cochlear half-turns I (Fig. 5a, b, Supplementary Fig. 3d). In addition, click-evoked (Fig. 5c) and tone burst-evoked (8 to 32 kHz) (Fig. 5d) auditory brainstem response (ABR) thresholds were considerably elevated in LPS-challenged *sgk1^{−/−}* mice than *sgk1^{+/+}* mice. To evaluate the vestibular function of the experimental model, the positive peak (P1) and negative peak (N1) latency in vestibular-evoked myogenic potentials (VEMPs) in response to clicks presented at 100 dB nHL were assessed. After administration of LPS, both *sgk1^{−/−}* and *sgk1^{+/+}* mice showed prolonged P1 and N1 latency (Fig. 5e). In addition, LPS-challenged *sgk1^{−/−}* mice showed more prolonged P1 latency compared to *sgk1^{+/+}* mice, but no differences in the N1 latency or amplitudes (Fig. 5e). We further subjected animals to rotarod tests and quantified their performance based on the time the animal remained on the rotating rod with increasing acceleration. After training, LPS-challenged *sgk1^{−/−}* mice showed a shorter latency time to fall from the accelerating rotarod than the *sgk1^{+/+}* mice (Fig. 5f and Supplementary Movie 1). Thus, SGK1 deficiency increases the severity of hydrops and

exacerbates damage to auditory and vestibular function in the LPS-induced EH mouse model.

Next, we examined the expression of inflammasome components in *sgk1^{−/−}* mice. The plasma concentrations of caspase-1 and IL-1β were significantly increased in *sgk1^{−/−}* mice after LPS challenge (Fig. 5g). Accompanying studies of inner ear tissue extracts subjected to western blotting revealed increased NLRP3 expression and cleavage of caspase-1 and IL-1β in extracts from *sgk1^{−/−}* mice than in those from *sgk1^{+/+}* mice (Fig. 5h). This suggests that SGK1 inhibits NLRP3 activity and suppresses activation of caspase-1 and IL-1β in vivo. TUNEL staining of the basilar membrane of *sgk1^{−/−}* mice suggested that SGK1 deficiency leads to increased apoptosis of inner ear hair cells (Fig. 5i). Taken together, these data support the idea that *sgk1^{−/−}* mice are more susceptible to LPS-induced audiovestibular dysfunction and inner ear inflammatory than are *sgk1^{+/+}* mice.

### Pharmacological inhibition of SGK1 increases the severity of LPS-induced EH and audiovestibular symptoms in vivo

The results of the above studies prompted us to explore whether a similar situation would occur in the case of SGK1 inhibition. We therefore p.a. injected a small-molecule compound GSK650394, which acts as a SGK1 inhibitor[40], to C57BL/6 mice administered with LPS. GSK650394-treated mice showed aggravated EH in cochlear half-turns I-IV (Fig. 6a, b), increased hearing damage (Fig. 6c, d), and exacerbation of vestibular dysfunction (Fig. 6e, f). Under inflammasome-inducing conditions, the expression of NLRP3, cleaved caspase-1 and IL-1β, and the plasma levels of caspase-1 and IL-1β were significantly higher in GSK650394-treated mice than in control groups (Fig. 6g, h). Furthermore, TUNEL staining of the basilar membrane showed inner ear hair cell damage in GSK650394-treated mice challenged with LPS (Fig. 6i). These data indicate that inhibition of SGK1 in mice deteriorates the disease severity in LPS-induced EH model, consistent with our observations in *sgk1^{−/−}* mice.

### Blockade of NLRP3 signaling ameliorates the severity of EH and audiovestibular symptoms in *sgk1^{−/−}* mice

Since SGK1 inhibition or deficiency is associated with increased NLRP3 production, we next investigated whether blockade of NLRP3 signaling could inhibit EH and audiovestibular symptoms induced in *sgk1^{−/−}* mice as demonstrated above. MCC950 is an effective inhibitor of NLRP3, which could interfere with NLRP3 oligomerization and potently impede NLRP3 activation[41]. Hence, we constructed an EH mouse model in *sgk1^{−/−}* mice treated with or without MCC950. As mentioned previously, *sgk1^{−/−}* mice had more severe EH and audiovestibular symptoms than *sgk1^{+/+}* mice, as evidenced by various parameters of vestibular and auditory function (Fig. 7a–f, Supplementary Movie 2) and by increased caspase-1 and IL-1β production in plasma or NLRP3, caspase-1 and IL-1β content of inner tissues (Fig. 7g, h) and number of apoptotic hair cells (Fig. 7i). In contrast, *sgk1^{−/−}* mice treated with MCC950 resulted in alleviation of EH (Fig. 7a, b), reversed the threshold increases in click-evoked ABR and tone burst-evoked ABR in the frequency range from 8 kHz to 32 kHz caused by SGK1 deficiency (Fig. 7c, d), attenuated the increase in VEMP P1 latency, and increased the residence time on the rotarod (Fig. 7f). Plasma concentrations of

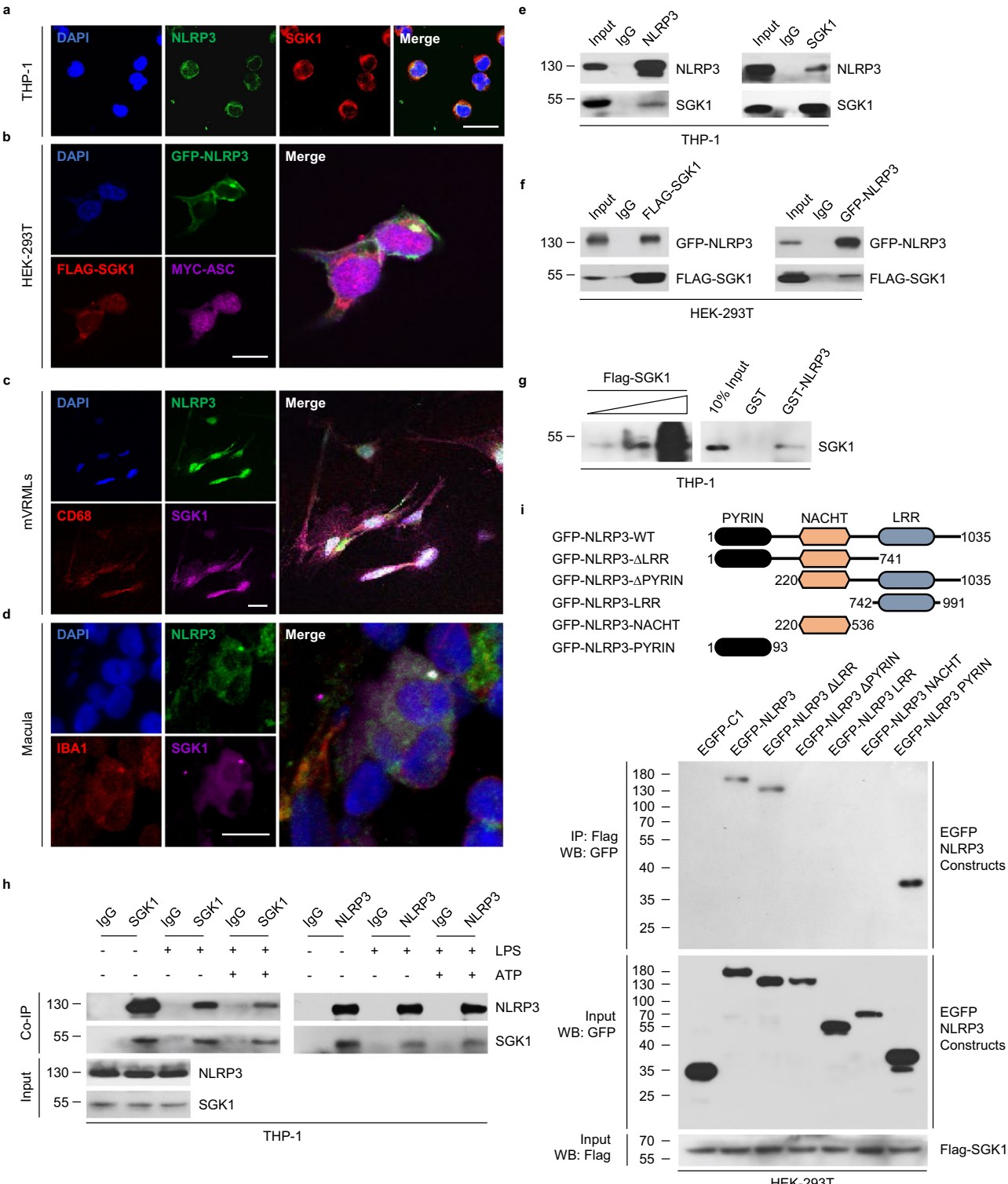

**Fig. 3 | SGK1 co-localizes and interacts with NLRP3. a** Immunofluorescence staining for DAPI (blue), NLRP3 (green), and SGK1 (red) in THP-1 cells that were stimulated with LPS (200 ng/mL, 3 h) and ATP (5 mM, 30 min), scale bar = 50 μm. **b** Representative confocal images of HEK-293T cells transiently expressing GFP-NLRP3 (green), FLAG-SGK1 (red), MYC-ASC (magenta), and staining DAPI (blue), which were stimulated with LPS (200 ng/mL, 3 h) and ATP (5 mM, 30 min), scale bar = 50 μm. **c** Immunofluorescence staining for DAPI (blue), NLRP3 (green), CD68 (red), and SGK1 (magenta) in primary mVRML cells of WT mice which were stimulated with LPS (200 ng/mL, 3 h) and ATP (5 mM, 30 min), scale bar = 10 μm. **d** Immunofluorescence staining for DAPI (blue), NLRP3 (green), IBA1 (red), and SGK1(magenta) in macula of MD patients, scale bar = 10 μm. **e** Co-IP analysis of the interaction of NLRP3

and SGK1 from the total lysates of THP-1 cells which were stimulated with LPS (200 ng/mL, 3 h) and ATP (5 mM, 30 min). **f** Co-IP analysis of the interaction between GFP-NLRP3 and FLAG-SGK1 from the total lysates of HEK-293T cells which were stimulated with LPS (200 ng/mL, 3 h) and ATP (5 mM, 30 min). **g** GST pull-down assay with GST-fused NLRP3 and in vitro translated SGK1. **h** Co-IP analysis of the interaction of NLRP3 and SGK1 from the total lysates of THP-1 cells which were stimulated with saline, LPS (200 ng/mL, 3 h) or LPS + ATP (5 mM, 30 min). **i** Domain-mapping experiment showing details of the NLRP3-SGK1 interaction in plasmid-transfected HEK-293T cells. mVRMLs, mouse vestibular-resident macrophage-like cells; Casp-1 caspase-1. Source data are provided as a Source Data file.

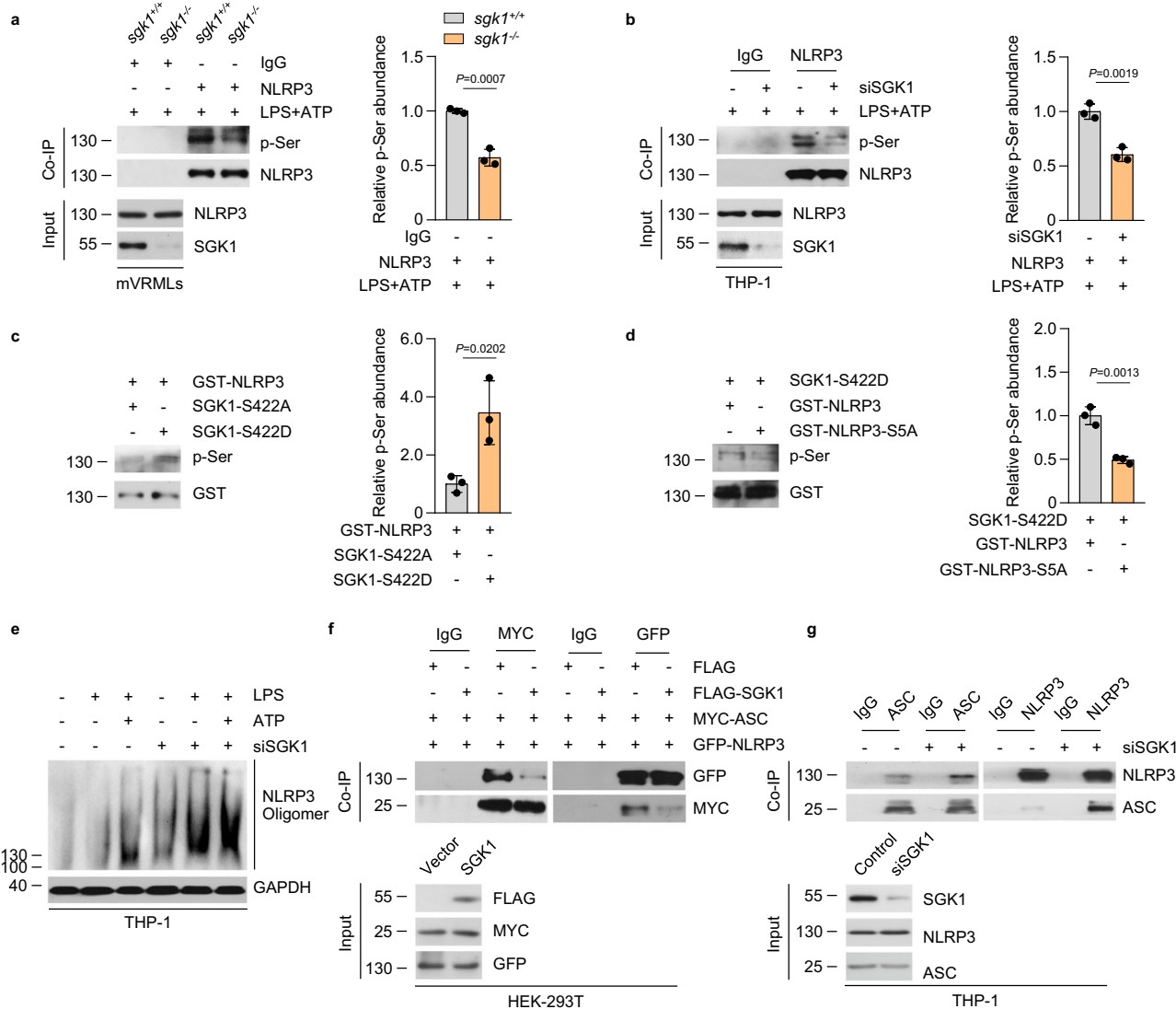

**Fig. 4 | SGK1 phosphorylates NLRP3 at serine 5 and inhibits inflammasome formation. a** Co-IP analysis of the serine phosphorylation of NLRP3 from the total lysates of primary mVRML cells of *sgk1^+/+* or *sgk1^−/−* mice upon stimulation with LPS (200 ng/mL, 3 h) and ATP (5 mM, 30 min). **b** Co-IP analysis of the serine phosphorylation of NLRP3 from the total lysates of control and *SGK1* siRNA-transfected THP-1 cells, which were stimulated with LPS (200 ng/mL, 3 h) and ATP (5 mM, 30 min). **c** In vitro kinase assay results showing the serine phosphorylation of NLRP3 after incubation with active SGK1-S422D or inactive SGK1-S422A via western blotting analysis. **d** In vitro kinase assay results showing the serine phosphorylation of wild-type NLRP3 or NLRP3-S5A after incubation with active SGK1-S422D via western blotting analysis. **a–d**, Protein expression was quantified by gray scanning, *n* = 3 biologically independent experiments. **a, b** NLRP3 served as a loading control

for western blotting. **c, d** GST served as a loading control for western blotting. **e** SDD-AGE assay showing NLRP3 oligomerization in control or *SGK1* siRNA-transfected THP-1 cells stimulated with saline, LPS (200 ng/mL, 3 h) or LPS + ATP (5 mM, 30 min). **f** Co-IP analysis of interaction between GFP-NLRP3 and MYC-ASC in the absence and presence of FLAG-SGK1 in HEK-293T cells stimulation with LPS (200 ng/mL, 3 h) and ATP (5 mM, 30 min). **g** Co-IP analysis of interaction between NLRP3 and ASC in control or *SGK1* siRNA-transfected THP-1 cells, which were stimulated with LPS (200 ng/mL, 3 h) and ATP (5 mM, 30 min). Results are presented as mean ± SD. Statistical analyses were carried out via two-sided *t*-test for **a–d**. mVRMLs, mouse vestibular-resident macrophage-like cells; Casp-1 caspase-1. Source data are provided as a Source Data file.

caspase-1 and IL-1β and the expression of NLRP3 and cleavage of caspase-1 and IL-1β in the inner ears of *sgk1^−/−* mice were significantly reduced in the MCC950-treated group (Fig. 7g, h). MCC950 treatment decreased positive TUNEL staining in *sgk1^−/−* mice (Fig. 7i). Together, these data suggest that inhibiting NLRP3 inflammasome activity might have therapeutic potential for inner ear damage induced by SGK1 deficiency-associated inflammatory disorders.

## Discussion

Studies on autoinflammatory disorders have expanded from rare monogenic diseases to complex polygenic autoinflammatory diseases such as adult-onset Still's disease, Crohn's disease, ulcerative colitis, or sarcoidosis[42,43]. Many of these autoinflammatory diseases are

characterized by high levels of the proinflammatory cytokine IL-1β and associated with NLRP3 inflammasome activation[44,45]. Emerging evidence suggests that NLRP3 may cause hearing loss by inducing local autoinflammation within the inner ear[9]. In some patients with MD, high levels of IL-1β and TNF-α might be suggestive of chronic inflammatory disorder[46]. However, previous studies have revealed neither the role of inflammatory activation in MD development nor its regulatory mechanisms. In summary, our results suggest a critical role of SGK1-NLRP3 interactions in VRML cells in autoinflammatory responses during MD pathogenesis. Under normal conditions, SGK1 binds to the PYD domain of NLRP3 and phosphorylates NLRP3 at S5, thereby preventing the assembly of the mature inflammasome. SGK1 deficiency augments NLRP3 inflammasome activation, which recruits ASC and

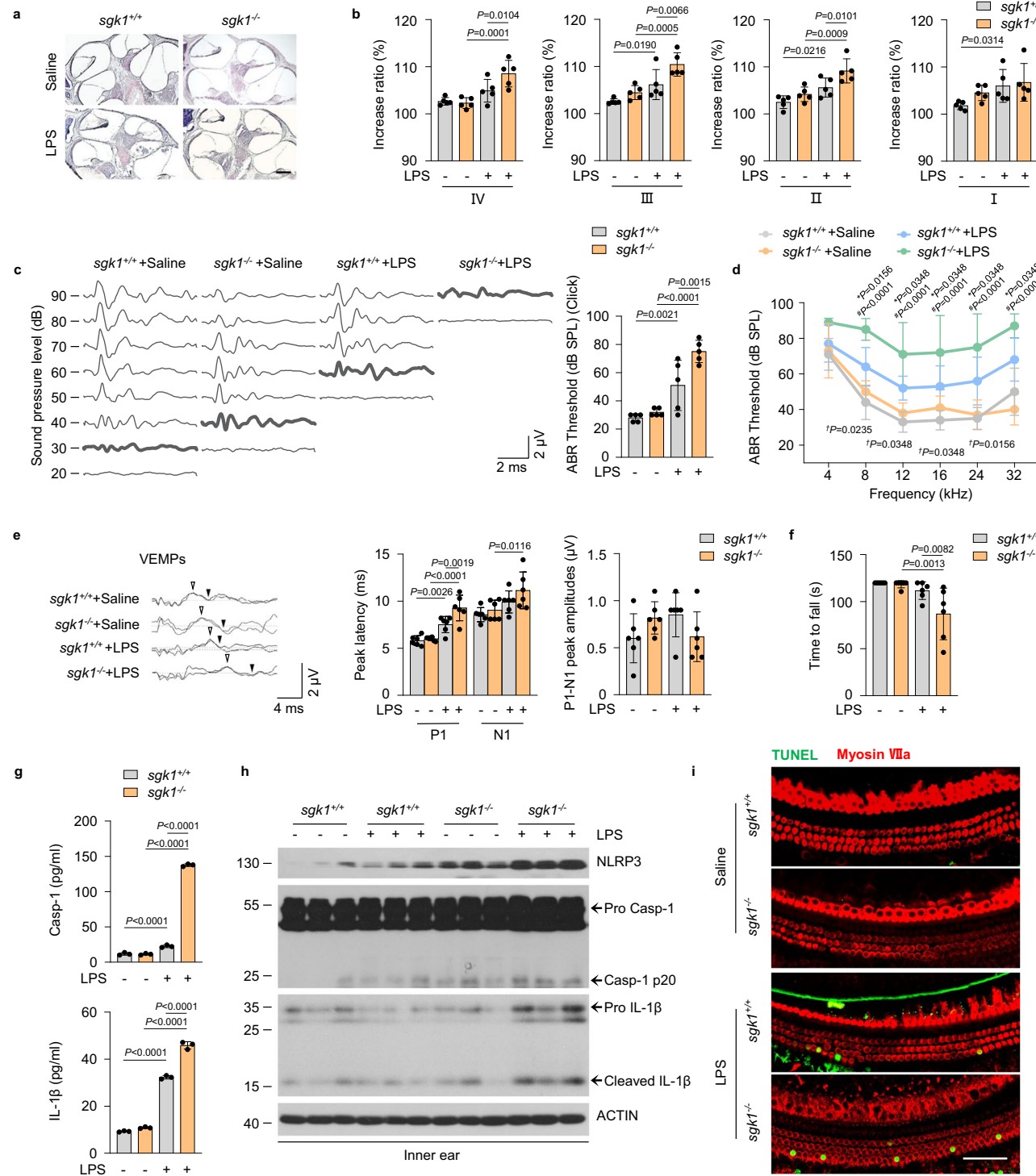

**Fig. 5 | SGK1 deficiency exacerbated the severity of LPS-induced EH and audiovestibular symptoms in vivo. a–i** *sgk1⁺/⁺* and *sgk1⁻/⁻* mice were treated with LPS or saline. **a** Representative images of mid-modiolar cochlear sections, scale bar = 200 μm. **b** Measurements of IR-L in cochlear half-turns I–IV (*n* = 5 mice per group). **c** Representative serial ABR wave recordings and thresholds in response to click sounds (*n* = 5 mice per group). **d** ABR thresholds in response to pure tone bursts across all frequencies tested (4, 8, 12, 16, 24, and 32 kHz) (*n* = 5 mice per group). *P indicates p-values for *sgk1⁺/⁺* +LPS versus *sgk1⁻/⁻*+LPS; #P indicates p-values for *sgk1⁻/⁻*+LPS versus *sgk1⁻/⁻*+Saline; †P indicates p-values for *sgk1⁺/⁺* +LPS versus *sgk1⁺/⁺* +Saline. **e** Representative click-evoked VEMPs waves, P1-N1 peak amplitudes, and the P1 (white triangle) and N1 (black triangle) peak latencies of VEMPs at 100 dB nHL

(*n* = 6 mice per group). **f** Quantification of rotarod test (*n* = 6 mice per group). **g** ELISA for plasma Casp-1 and IL-1β levels (*n* = 3 mice per group). **h** Western blot analysis showing NLRP3, SGK1, pro Casp1, Casp-1 p20, pro IL-1β and cleaved IL-1β levels in inner ears (*n* = 3 mice per group). **i** Immunofluorescence staining for TUNEL (green) and Myosin VIIa (red) in the basilar membrane, scale bar = 50 μm. Results are presented as mean ± SD. Statistical analyses were carried out via one-way ANOVA for **b**, **c**, **e**–**g**, and two-way ANOVA for **d**. EH endolymphatic hydrops, IR-L Increase ratios (IR) of the length of the Reissner's membrane, ABR auditory brainstem response, VEMPs vestibular-evoked myogenic potentials, P1 positive peak, N1 negative peak, Casp-1 caspase-1. Source data are provided as a Source Data file.

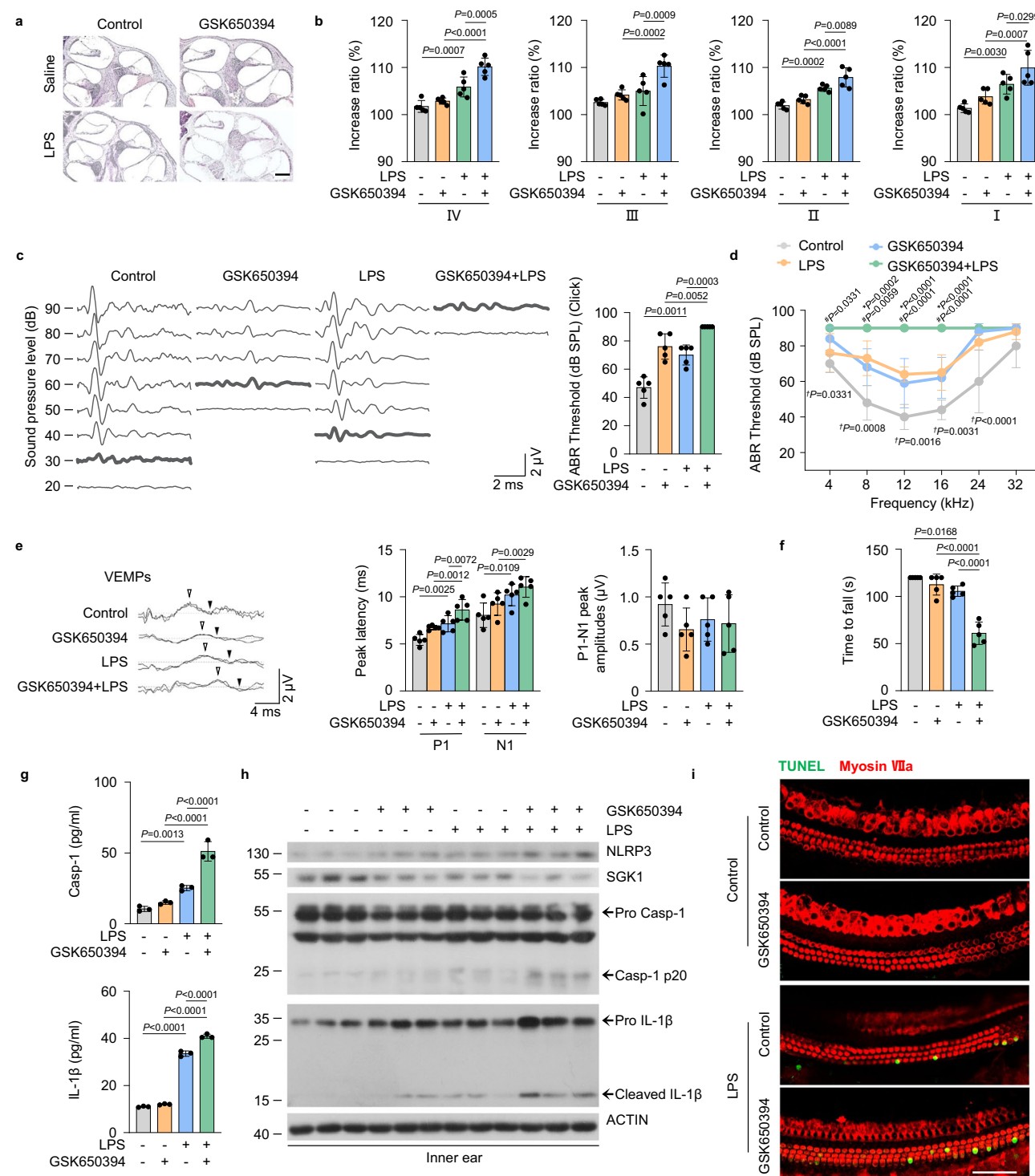

**Fig. 6 | Pharmacological inhibition of SGK1 increases the severity of LPS-induced EH and audiovestibular symptoms in vivo. a–i** The WT mice were untreated or pretreated with GSK650394 and challenged with LPS or saline. **a** Representative images of mid-modiolar cochlear sections, scale bar = 200 μm. **b** Measurements of IR-L in cochlear half-turns I–IV (*n* = 5 mice per group). **c** Representative serial ABR wave recordings and thresholds in response to click sounds (*n* = 5 mice per group). **d** ABR thresholds in response to pure tone burst across all frequencies tested (4, 8, 12, 16, 24, and 32 kHz) (*n* = 5 mice per group). *P indicates p-values for GSK650394 + LPS versus LPS; #P indicates p-values for GSK650394 + LPS versus GSK650394; †P indicates p-values for LPS versus Control. **e** Representative click-evoked VEMPs waves, P1-N1 peak amplitudes, and the P1

(white triangle) and N1 (black triangle) peak latencies of VEMPs at 100 dB nHL (*n* = 5 mice per group). **f** Quantification of rotarod test (*n* = 5 mice per group). **g** ELISA for plasma Casp-1 and IL-1β levels (*n* = 3 mice per group). **h** Western blot analysis showing NLRP3, SGK1, pro Casp1, Casp-1 p20, pro IL-1β and cleaved IL-1β levels in inner ears (*n* = 3 mice per group). **i** Immunofluorescence staining for TUNEL (green) and Myosin VIIa (red) in the basilar membrane, scale bar = 50 μm. Results are presented as mean ± SD. Statistical analyses were carried out via one-way ANOVA for **b**, **c**, **e**–**g**, and two-way ANOVA for **d**. EH endolymphatic hydrops, IR-L Increase ratios (IR) of the length of the Reissner's membrane, ABR auditory brainstem response, VEMPs vestibular-evoked myogenic potentials, P1 positive peak, N1 negative peak, Casp-1 caspase-1. Source data are provided as a Source Data file.

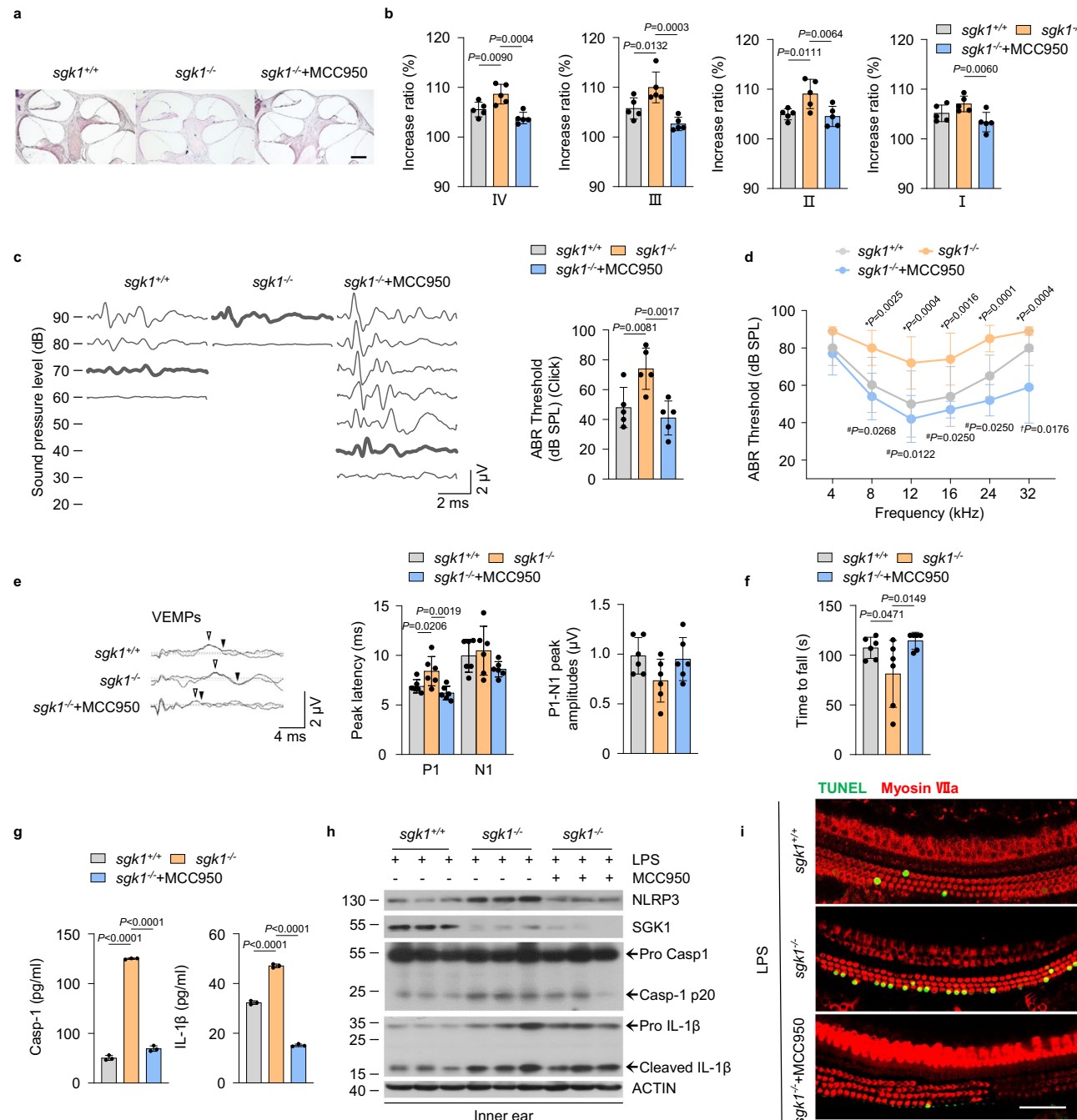

**Fig. 7 | Blockade of NLRP3 signaling ameliorates EH and audiovestibular symptoms in *sgk1⁻/⁻* mice. a–i** The *sgk1⁺/⁺* and *sgk1⁻/⁻* mice were treated with saline or MCC950, and followed by LPS stimulation. **a** Representative images of EH severity in mid-modiolar cochlear sections, scale bar = 200 μm. **b** Measurements of IR-L in cochlear half-turns I–IV (*n* = 5 mice per group). **c** Representative serial ABR wave recordings and thresholds in response to click sounds (*n* = 5 mice per group). **d** ABR thresholds in response to pure tone burst across all frequencies tested (4, 8, 12, 16, 24, and 32 kHz) (*n* = 5 mice per group). *P indicates *p*-values for *sgk1⁻/⁻*+LPS + MCC950 versus *sgk1⁻/⁻*+LPS; #P indicates *p*-values for *sgk1⁺/⁺*+LPS versus *sgk1⁻/⁻*+LPS; †P indicates *p*-values for *sgk1⁺/⁺* +LPS versus *sgk1⁻/⁻*+LPS + MCC950. **e** Representative click-evoked VEMP waves, P1-N1 peak amplitudes, and the P1 (white triangle) and N1 (black triangle) peak latencies of VEMPs at 100 dB nHL

(*n* = 6 mice per group). **f** Quantification of rotarod test (*n* = 6 mice per group). **g** Concentrations of Casp-1 and IL-1β in plasma of treated mice as measured via ELISA (*n* = 3 mice per group). **h** Western blot analysis showing NLRP3, SGK1, pro Casp1, Casp-1 p20, pro IL-1β, and cleaved IL-1β levels in the ears (*n* = 3 mice per group). **i** Immunofluorescence staining for TUNEL (green) and Myosin VIIa (red) in the basilar membrane, scale bar = 50 μm. Results are presented as mean ± SD. Statistical analyses were carried out via one-way ANOVA for **b**, **c**, **e**–**g**, and two-way ANOVA for **d**. EH endolymphatic hydrops, IR-L Increase ratios (IR) of the length of the Reissner's membrane, ABR auditory brainstem response, VEMPs vestibular-evoked myogenic potentials, P1 positive peak, N1 negative peak, Casp-1 caspase-1. Source data are provided as a Source Data file.

pro caspase-1 to assemble the mature inflammasome. These results in the maturation and secretion of IL-1β and induction of pyroptosis, thereby facilitating the severity of LPS-induced EH and audiovestibular dysfunction (Fig. 8).

SGK1, belonging to a subfamily of serine/threonine kinases called AGC protein kinases[47], is involved in multiple cell signaling pathways and phosphorylation cascades and plays an important role in ion channel function, inflammation, and cell survival. SGK1 is expressed in

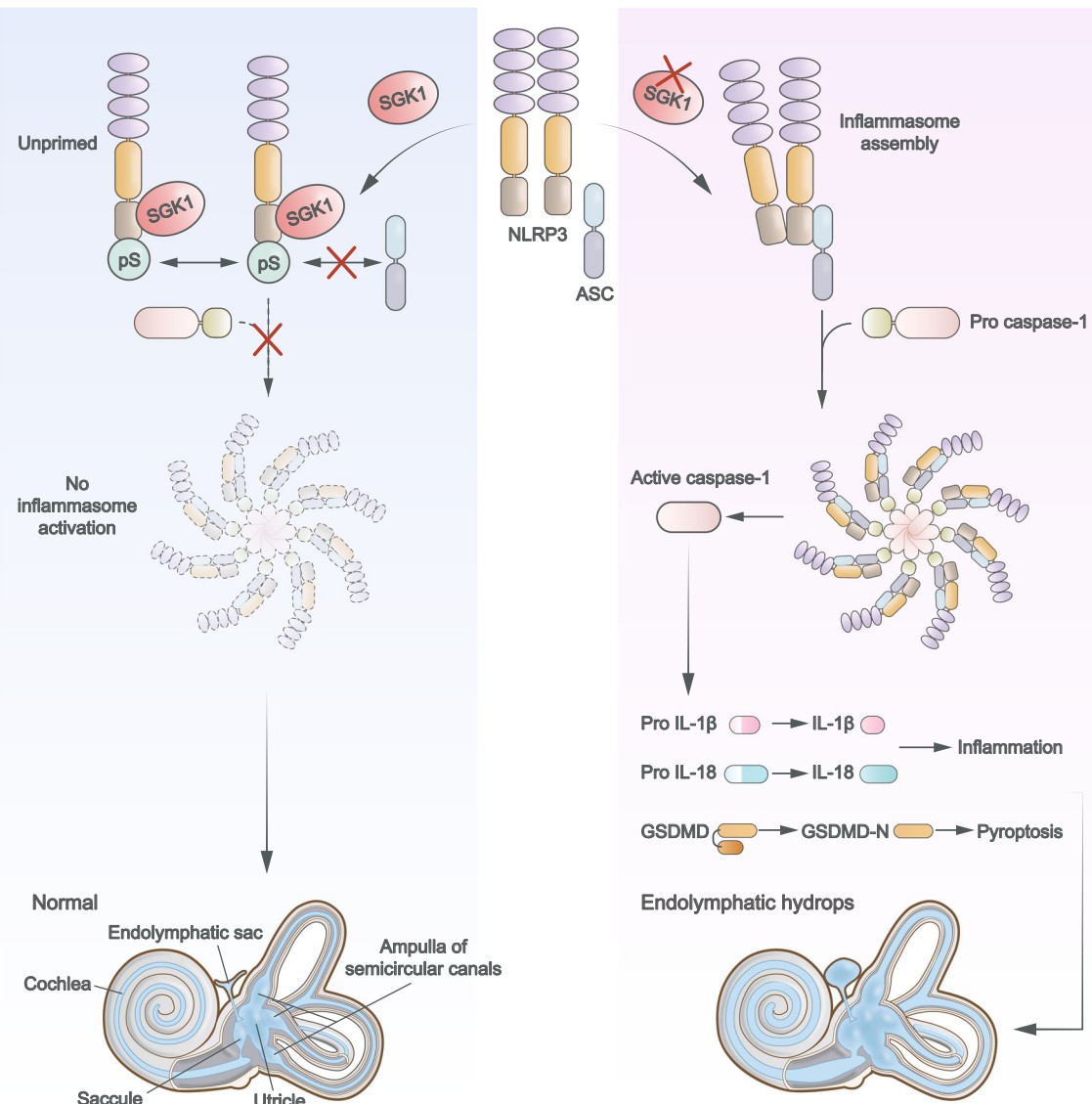

**Fig. 8 | Schematic model for the mechanism by which SGK1 inhibits NLRP3 inflammasome activation to control inner ear inflammation.** Under normal conditions, SGK1 binds to the PYD domain of NLRP3 and phosphorylates NLRP3 at S5, thereby preventing the assembly of the mature inflammasome. SGK1 deficiency then augments NLRP3 inflammasome activation, which recruits ASC and pro caspase-1 to assemble the mature inflammasome, resulting in the maturation and secretion of IL-1β and IL-18 and induction of pyroptosis, thereby facilitating the severity of LPS-induced EH.

the stria vascularis, spiral ganglion neurons, spiral limbus, organ of Corti, and the spiral ligament of rat and guinea pig cochlea[24,48]. We showed that SGK1 levels in the VEO of MD patients were lower than those in AN patients. Reduction in SGK1 expression was also observed in LPS-treated macrophages and mice with EH. Consistent with our findings, previous studies have reported that SGK1 expression decreased after treatment with LPS and other activators of the nuclear factor (NF)-κB pathway, such as IL-1β and TNF-α, in renal collecting duct principal cells[29]. SGK1 activity and expression are downregulated in the mid-secretory endometrial stromal cells of women suffering from recurrent pregnancy loss[49,50], lung tissues of cecal ligation and puncture (CLP)-induced acute lung injury[51], and colonic tracts of patients with ulcerative colitis[52], which sensitizes these patients to inflammation. Moreover, SGK1 is a primary regulator of ion channels and the gene encoding this protein is an important anti-apoptotic gene[53,54]. Reduction in SGK1 expression in the VEO of patients with MD may explain some of the features associated with MD, such as the dysregulation of sodium absorption, resulting EH, and increased hair cell loss. However, reasons behind the levels of SGK1 being reduced in

the VEO of patients with MD compared to those in patients with AN remain to be determined. A previous study revealed that the activation of NF-κB pathway downregulates SGK1 expression by inhibiting SGK1 gene transcription via the binding of the NF-κB p65 subunit to specific sites of the SGK1 promoter[29]. Furthermore, the variant rs4947296 regulates gene expression in the TWEAK/Fn14 pathway, which induces NF-κB translation, which is in turn associated with abnormal inflammatory response in MD[55]. We also observed increased mRNA expression of NF-κB in VRML cells from patients with MD. Therefore, SGK1 downregulation in MD may be attributed to the activation of NF-κB via the TWEAK/Fn14 pathway associated with the allelic variant rs4947296. Further investigation will provide greater insight into the physiological cues of SGK1 downregulation and role of SGK1 in MD.

Proinflammatory cytokines, such as interleukin-1β (IL-1β), have been implicated in a wide variety of middle and inner ear conditions, including MD[12,56,57]. Consistently, we found that an increasement of IL-1β both in plasma and VEO of MD patients. Of note, our finding that IL-1β levels are elevated in 75% patients with MD of active phase, but 25% patients with MD of quiescent phase, which suggests that vertigo

episodes may lead to increased IL-1β levels. Earlier studies have proposed that IL-1β activates NF-κB resulting in the transcriptional activation of several genes, including inflammatory mediators[58,59]. Thus, NF-κB may negatively regulates SGK1, leading to MD pathogenesis through positive regulatory loops with NLRP3 and IL-1β. In addition to immune factors, viral infections and autoimmune or structural dysfunction of the inner ear have been reported as causes of MD[60]. IL-1β expression levels are elevated during bacterial or fungal infections[61–63], or inflammation in patients with active autoimmune diseases[64,65]. This indicates that, in addition to this SGK1-dependent mechanism, there may be other initial triggers of IL-1β elevation and lead to the progression of MD.

Generally, activation of macrophage NLRP3 pyrosomes occurs in the cytoplasm. In current study, we found that NLRP3 expressed in both nucleus and cytoplasm in mVRML cells. This unusual nuclear localization of NLRP3 has also been observed in some other cells[66–68]. We speculate a difference in NLRP3 localization across cell lines according to the tissue heterogeneity, which may indicate that in mVRML cells, the cytoplasmic localization of NLRP3 may promote the assembly of inflammatory bodies, whereas its nuclear localization may be beneficial to the transcription function of inflammatory bodies.

Accumulating evidence suggests that proinflammatory cytokines secreted by macrophages may predispose the inner ear microenvironment to immune maladaptation[10,11]. NLRP3 phosphorylation is linked to the induction or inhibition of inflammasome activity[69] and several kinases modify NLRP3. JNK1 positively regulates NLRP3 assembly by phosphorylating it at S194[70], whereas Akt negatively regulates NLRP3 assembly by phosphorylating it at S5[36]. SGK1 is a multifunctional PI3-kinase-dependent serine-threonine kinase structurally homologous to Akt that mediates phosphorylation of several proteins[71–75]. Mechanistically, we showed that SGK1 directly phosphorylates NLRP3 at S5. In an unstimulated state, SGK1 phosphorylates the PYD domain of NLRP3, which hinders the interaction of NLRP3 PYDs with ASC PYDs via electrostatic repulsion, thereby preventing accidental NLRP3 assembly[38]. The interaction between SGK1 and NLRP3 is weakened in the presence of inflammatory signals, leading to disappearance of the physiological gatekeeper, followed by NLRP3 oligomerization and promotion of the interaction between NLRP3 and ASC.

While this study was in progress, a potential regulatory role of SGK1 in the NLRP3 inflammasome was proposed by Kwon et al.[76], Clauzure et al.[77] and Gan et al.[78], who showed that silencing or pharmacological inhibition of SGK1 using EMD638683 or GSK650394 may abrogate IL-1β secretion in response to stimulation by LPS + ATP, Cl−, or angiotensin II via suppressing NLRP3 inflammasome activity. However, our findings regarding the effects of SGK1 inhibitors on NLRP3 activation differ from those of previous studies. Some studies have reported that SGK1 stimulates inflammatory activation[79,80], while others found an inhibitory role for SGK1[49,81]. Previous reports have shown that SGK1 deficiency enhances the production of interferon-gamma and Th1 cell-mediated immunity upon viral infection[82]. Here, we propose a direct role of SGK1 in suppressing inner ear inflammation by blocking the inflammasome components.

SGK1 negatively regulates IL-1β expression by suppressing NLRP3 inflammasome activation. This is supported by the fact that knockdown or pharmacological inhibition of SGK1 expression in the whole animal renders them susceptible to increased LPS-induced EH and vestibular dysfunction, which is accompanied by NLRP3 inflammasome activity and IL-1β production. In addition, all symptoms of LPS-induced EH and audiovestibular dysfunction in *sgk1*-knockout mice were reversed in the presence of MCC950, which is the most potent and specific small-molecule inhibitor of NLRP3 activation known.

Previous studies have proposed that repeated exposure of hair cells to toxic levels of K+-enriched perilymph, overpressure, and sudden rupture of distended membranes explain the long-term vestibular and auditory damage in MD[83]. We showed that SGK1-deficient mice had significantly more severe vestibular hair cell impairments than WT mice. In addition, IL-1β directly damaged cochlear and vestibular hair cells and vestibular nerves. These results demonstrated that NLRP3 inflammasome activation, accompanied by elevated IL-1β production, was responsible for the auditory and vestibular dysfunction.

Collectively, our study suggests that SGK1 as a negative regulator of the NLRP3 inflammasome and link to macrophage autoinflammation in a murine EH model. These findings further expand our understanding of the innate immune pathways of the inner ear. Furthermore, our findings provide convincing evidence that targeting the SGK1-NLRP3-IL-1β axis may be useful for treating MD.

## Methods
All experiments performed in this study were compiled with ethical regulations and approved by the Animal Care Committee of Shandong University (Jinan, China) and the Ethics Committee of Shandong Provincial ENT Hospital affiliated to Shandong University. The details are described in the respective sections below.

### Animal studies
Wild-type (WT) C57BL/6 (2-month-old male) mice were purchased from the Animal Center of Shandong University, China. *Sgk1* knocked out (*sgk1−/−*, male, 2 months) mice were purchased from Cyagen Biosciences. Mice were housed in a temperature-controlled (20–22 °C) room with 40–70% humidity, subjected to a 12/12 h light/dark cycle, and had free access to food and drinking water. Experiments were performed on age- and sex-matched 8- to 10-week-old mice weighing 17-25 g. All study protocols were approved by the Animal Care Committee of Shandong University, China (No. ECAESDUSM 20,123,011) and conformed with the Guideline for the Care and Use of Laboratory Animal of the National institutes of Health.

### LPS-induced EH
Mice were challenged with LPS (2 mg/kg, Sigma, Supplementary Table 1) dissolved in saline through postauricular (p.a.) injection[84] once a day for 3 days to establish the EH mouse model. The control groups were injected p.a. with an equivalent of 0.9% physiological saline.

To evaluate the effect of SGK1 inhibition, we treated the EH mice model with SGK1 inhibitor GSK650394 (10 mM, 40 µL/d/mouse, p.a., MCE)[85]. According to the instructions, GSK650394 was dissolved in dimethyl sulfoxide (DMSO, Sigma) and the corresponding control group mice were treated with an equal amount of DMSO.

To evaluate the roles of NLRP3 in LPS-induced EH, *sgk1−/−* mice were intraperitoneally (i.p.) injected with NLRP3 inhibitor MCC950 (10 mg/kg, AdipoGen) or saline. MCC950 or saline was injected for 3 consecutive days before LPS challenge[86].

In all cases, hearing and vestibular function were tested on day 5. Thereafter, the mice were euthanized, blood samples were collected in a tube containing ethylenediaminetetraacetic acid, plasma was separated immediately, and inner ears were subjected to protein or mRNA expression and EH analysis.

### Human studies
Forty patients aged 34–87 years (average age 59.9 years) at Shandong Provincial ENT Hospital affiliated to Shandong University from January 2019 to December 2021 were enrolled in this study (clinical information of patients with MD is listed in Supplementary Table 2). These patients had been diagnosed with MD according to the 2015 Diagnostic criteria for Menière's disease[1]. Of these patients, 20 were all hospitalized, and their vertigo was not effectively controlled; hence, they experienced frequent episodes. Their peripheral blood was collected during the active phase of MD. Another 20 of these patients were free from vertigo attack for at least 3 months, and their peripheral blood samples were collected during the quiescent phase[87]. Patients

with a history of thermal burns, Alzheimer's disease, cancer, or recent influenza, fever, were excluded. The control group consisted of 20 healthy volunteers and 9 patients with AN aged 47–68 years (average age 55.7 years) with no history of sudden vertigo from the same age group, living in the same region, and possessing socioeconomic indicators similar to those of the patient group. The subjects did not suffer from any systemic disease. The gender distribution and mean age of the patients and controls were not significantly different ($P > 0.05$ for the chi-square test and $t$-test).

Patients with MD underwent labyrinthectomy for their intractable disease and the AN group underwent tumor removal via the translabyrinthine approach. The ampulla and macula were sampled during surgery and the tissues were immediately frozen in liquid nitrogen. Peripheral blood was sampled from all participants in each group and were stored in a tube containing ethylenediaminetetraacetic acid. Peripheral blood mononuclear cells (PBMCs) and plasma were immediately separated and stored at −80 °C.

The study was carried out according to the principles of the Declaration of Helsinki revised in 2013 for investigation with humans. The Ethics Committee of Shandong Provincial ENT Hospital affiliated to Shandong University approved this study (XYK20181211). Informed consent was obtained from all participants.

## Cell lines and culture

All the cell lines were obtained from Procell Life Science & Technology. HEK-293T cells (Procell, CL-0005) were maintained in Dulbecco's Modified Eagle's Medium (DMEM, Gibco) supplemented with 10% fetal bovine serum (FBS, Gibco) and penicillin/streptomycin (Gibco, 10,000 units/ml). THP-1 cells (Procell, CL-0233) were grown in Roswell Park Memorial Institute (RPMI)−1640 (Gibco) media supplemented with 10% FBS, β-mercaptoethanol (Procell, 0.05 mM), and penicillin/streptomycin. Cells were maintained in a humidified incubator equilibrated with 5% $CO_2$ at 37 °C. Transfections were performed using Lipofectamine® RNAiMAX reagent (Invitrogen) according to the manufacturer's instructions. HEK-293T cells for the overexpression experiment were transfected using the calcium phosphate method as per the manufacturer's instructions (Polyplus Transfection). In these experiments, cells were stimulated with LPS (200 ng/ml) for 3 h and/or ATP (5 mM) for 30 min, unless otherwise indicated. The treatment with z-VAD-FMK (20 μM) or Ac-YVAD-cmk (40 μM) was performed 30 min earlier than stimulation with LPS.

## Tissue and primary cell culture

Temporal bones and cochleae were collected from Postnatal Day 3 (P3) C57BL/6 mice and placed in cold sterile Hank's balanced salt solution. The cochlear sensory epithelia, VEO, and vestibular ganglion were isolated and cultured in DMEM/F12 containing 10% FBS and 100 μg/mL ampicillin. Primary VRML cells from the VEO of mice were collected as described previously[88]. Briefly, freshly isolated VEO were rapidly cut into small pieces, which were cultured in melanocyte medium (MelM) with 1% melanocyte growth supplement (MelM) at 37 °C in the presence of 5% $CO_2$ for 3 days. The VRML cells were expanded, further selected, and validated based on CD68 expression. In these experiments, cells were stimulated with LPS (200 ng/ml) for 3 h and/or ATP (5 mM) for 30 min, unless otherwise indicated. Tissues were stimulated with IL-1β (10 ng/mL) for 24 h.

## IF

The frozen sections were permeabilized in 0.5% Triton X-100 for 1 h and immunoblocked with 5% bovine serum albumin (BSA) for an additional 1 h. For cells, approximately $10^5$ cells were plated on a coverslip. Treated cells were fixed in 4% PFA for 10 min, permeabilized with 0.1% Triton X-100 for 10 min, followed by blocking with 1% BSA for 30 min at room temperature. The specimens were incubated with different primary antibodies as listed in Supplementary Table 3 at 4 °C

overnight. The next day, cells were incubated with secondary antibodies (Invitrogen, 1:1,000) at room temperature for 1 h. Coverslips were then mounted with Prolong Gold with DAPI (Invitrogen) and the samples were observed under a laser scanning confocal microscope (Leica SP8; Leica, Germany).

## IHC

A DAB Detection Kit (Streptavidin-Biotin) kit was applied to examine the expression of the OCT-embedded vestibular apparatus tissue. Briefly, the endogenous peroxidase activity was blocked by incubation for 30 min in 0.3% $H_2O_2$. Nonspecific binding was blocked by a 20 min incubation with 10% normal goat serum. The sections were stained with different primary antibodies as listed in Supplementary Table 3 overnight at 4 °C. After rinsing with PBS and incubation with biotinylated anti-rabbit or anti-mouse IgG for 30 min at a dilution to 5 μg/ml, the sections were rinsed again with PBS and incubated with DAB complex for 1 min. The sections were counter-stained with hematoxylin, dehydrated, and omitted with Neutral balsam. Images were obtained using a Leica microscope at 40 × 10 magnification.

To evaluate the intensity of immunohistochemical reaction quantitatively, digital images were obtained and analyzed using a public domain ImageJ software. Intensity measurements were represented as the number in a 256-gray scale. Optical density values were corrected by subtracting the average values of background noise (mean background intensity) obtained from five image inputs. The optical density was then standardized by setting the threshold (mean background intensity) levels. Manipulation of the images was restricted to threshold and brightness adjustments to the whole image. At least 3 separate measurements per group were subjected to image analysis.

## qPCR

The total RNA of the mice cochleae and cell lines was extracted using Trizol reagent according to manufacturer's protocols (Invitrogen); 1 mg of RNA was used for reverse transcription using a high-capacity DNA reverse transcription kit. The relative expression levels of target gene mRNA were measured using qRT-PCR using an Eppendorf AG 22331 PCR machine (Hamburg, Germany) and qRT-PCR was performed using TaqMan master mix or Power SYBR green PCR master mix according to manufacturer's protocols on Step One Real-Time PCR system (Applied Biosystems). The sequences of primers used in this experiment are listed in Supplementary Table 4. mRNA expression profiles were normalized to levels of the housekeeping gene glyceraldehyde 3-phosphate dehydrogenase (*GAPDH*) or *ACTB* in each sample.

## ELISA

Paired antibodies and standard recombinant human IL-1β (RD), IL-18 (Abcam), caspase-1 (Abcam) and mouse IL-1β (Abcam) and caspase-1 (Novus) were used to determine cytokine concentrations according to manufacturer's instructions. A microplate reader (Bio-Rad) was used to detect the signals at 450 nm.

## Western Blot

The total protein of cochleae and cell lines was extracted with cold RIPA lysis buffer plus protease inhibitor cocktail for 30 min at 4 °C and then centrifuged at $12,000 \times g$ for 20 min at 4 °C. Protein concentrations were detected by the BCA Protein Assay Kit. The same amount of protein sample was denatured in 99 °C for 10 min and separated using 10% SDS-PAGE. Then, the proteins were transferred to polyvinylidene difluoride membranes (Millipore). The membranes were blocked in 5% skim milk for 1 h at room temperature and then were incubated with different primary antibodies as listed in Supplementary Table 5. The next day, the membranes were incubated with HRP-conjugated secondary antibodies purchased from Jackson (1:10,000) at room

temperature for 1 h. Finally, the protein signals were detected using an ECL kit (Millipore) and analyzed using ImageJ software. All blocking, incubation, and washing procedures were performed in TBST solution (Tris-buffered saline and 0.05% Tween 20).

## Flow cytometry

Cell apoptosis was measured using an Annexin V-FITC Apoptosis Detection Kit (Beyotime). Cells were harvested, washed with PBS solution, and resuspended with 195 µl binding buffer softly; 5 µl Annexin V-FITC and 10 µl PI were added and incubated in a dark room for 15 min. 30,000 cells of each group were measured by a LSRll flow cytometer and BD FACSDiva software version 9 (BD Bioscience) and analyzed with FlowJo software (FlowJo, LLC). Cells were dicscriminated from debris and clumps using the FSC-A/SCC-A gating strategy based on experience. Only single cells were used by using an FSC-H/FSC-A gating strategy and selecting cells along the diagonal. The gating strategy is provided in Supplementary Fig. 2e.

## Co-immunoprecipitation (Co-IP)

For immunoprecipitation assays, the cells were washed twice with cold PBS and the extracts were prepared by incubating the cells in lysis buffer (50 mM Tris–HCl; pH 7.4, 150 mM NaCl, 1 mM EDTA, 0.5% NP-40, 0.25% sodium deoxycholate, and protease-inhibitor cocktail) for 30 min at 4 °C and then centrifuging at $12,000 \times g$ for 15 min. Next, 500 µg protein samples were incubated with appropriate primary antibodies or normal rabbit/mouse IgG at 4 °C for 14 h with constant rotation, and then mixed with protein G Dynabeads (Invitrogen) for 2 h at 4 °C. After washing the beads thrice with the cell lysis buffer, the captured immune complexes were subjected to SDS-PAGE.

## GST pull-down assay

GST fusion proteins were expressed in BL21 (Vazyme) *Escherichia coli*, followed by ultrasonic disruption and purification using glutathione Sepharose 4B beads (GE). In vitro transcription and translation experiments were performed using rabbit reticulocyte lysate (TNT systems, Promega) according to the manufacturer's recommendation. The beads were co-incubated with the in vitro translated proteins and then washed with binding buffer. The eluates were then analyzed using SDS-PAGE.

## In vitro kinase assay

Recombinant GST-NLRP3 or GST-NLRP3-S3A was used as a substrate and incubated with recombinant SGK1 (S422A) or SGK1 (S422D) in kinase assay buffer containing 25 mM MOPS pH 7.2, 12.5 mM β-glycerophosphate, 25 mM MgCl₂, 5 mM EGTA, 2 mM EDTA and 0.25 mM DTT at 30 °C for 30 min. Reactions were stopped by adding SDS-PAGE sample buffer. The phosphorylate status were analyzed using SDS-PAGE.

## Identification of NLRP3 oligomerization

The total protein of cell lines was extracted with cold SDD-AGE buffer plus protease inhibitor cocktail for 30 min at 4 °C and then centrifuged at $12,000 \times g$ for 20 min at 4 °C. After protein concentration detection, same amounts of protein were separated by gels with 1% (w/v) agarose and 0.1% SDS without denaturation. Proteins were transferred to polyvinylidene difluoride membranes. The membranes were subjected to blocking, incubation, and detection as mentioned before.

## Auditory function evaluation

The auditory function in mice was evaluated using ABR. These responses were measured with a click stimulus and tone burst stimuli at 4, 8, 12, 16, 24, and 32 kHz using a TDT system 3 (Tucker-Davis Technologies) with 1,024 stimulus repetitions per record in a sound isolation booth. Mice were anesthetized with xylazine (10 mg/kg) and ketamine (100 mg/kg) via i.p. injection. Needle electrodes were inserted into the subcutaneous tissue at the vertex (recording electrode), the infra-auricular mastoid region of ipsilateral ear (reference electrode), and the back (ground electrode). The sound level started from a 90-dB sound pressure level (SPL) and sequentially decreased by 5-dB to the acoustic threshold. The ABR threshold was determined at each frequency, which refers to the minimal SPL resulting in a reliable ABR recording with one or more distinguishable waves clearly identified by visual inspection. The process was repeated for low SPLs around the threshold to ensure the waveform consistency. Following ABR hearing measurements, the cochleae were used for histological, biochemical, and molecular analyses.

## VEMPs

Click-evoked VEMP recordings from all animals were initiated with simultaneous recording of electromyography (EMG) potentials after the induction of anesthesia. A custom-made holder previously reported[89] was used for these recordings. In these experimental mice, the neck was hyperextended and stabilized with suspension wire fixed behind the front teeth. Mice, with elevated head and free legs, were kept in prone position. During recording, a platinum needle electrode was inserted into the neck extensors and the reference electrode was planted on the cervico-occipital region at the midline. The ground electrode was placed on the back. Each animal was subjected to VEMP testing with stimulus intensity of 100 dB nHL. The response threshold was the lowest threshold for the appearance of a waveform, with repeatability verified by consecutive runs (>3 times). Finally, the latencies and amplitudes of the negative and positive peaks were measured.

## Rotarod test

Animals were placed on a motorized rotating rod (ZH-600 B, Huaibei Zhenghua Biological Instruments Co., Ltd.) set at a maximum speed of 30 rpm. The rotation speed was gradually accelerated to 30 rpm for 2 min. Each mouse was trained 2 times per day for three days and tested twice. Thereafter, the average time to fall off the rotarod in 2 trials was measured for analysis.

## Cell counting

For IBA1-positive cell quantification, we imaged the VEO using a 400× objective and ImageJ software (National Institutes of Health). Briefly, a grid was overlaid on the tissue area of interest, and the "point selection" tool in ImageJ was used to manually count (and classify) cells according to their IBA1 expression. Only cells with a DAPI-stained nucleus were counted. Data were expressed as the average number of positive cells per millimeter squared of tissue section, and at least three tissue sections per group were analyzed.

## Quantitative assessment of changes in endolymphatic space in cochlea

Frozen sections of mouse inner ear were stained using an HE differentiation solution, gradient dehydrated with ethanol, and sealed with neutral gum. For quantitative assessment of the changes in the endolymphatic space, the length of the stretched Reissner's membrane (L) and the ideal length of the Reissner's membrane (L*) were measured in each cochlea as stated in the literature[90]. The increase ratios of the length change of the Reissner's membrane (IR-L = L/L* × 100%) were calculated to evaluate the EH level for upper and lower turns. The values were evaluated using ImageJ software.

## Statistical and reproducibility

All statistical analysis was performed with SPSS 13.0 software (SPSS Inc., USA). Data are shown as the mean ± standard deviation (SD). Two-side unpaired Student's *t*-tests, one-way analysis of variance (ANOVA) followed by Fisher's LSD post hoc test and two-way ANOVA followed by Tukey's test was performed. Differences with a *P*-value <0.05 were

considered to be statistically. Statistical details of analyses can be found in the figure legends. Mice were randomly divided to different groups in animal studies. No samples or animals were excluded from the analysis. No statistical method was used to predetermine the sample size. All samples were processed in blind fashion. The number of independent samples of each experiment can be found in the relevant figure legend. For western blots, qPCR, flow cytometry and other quantitative experiment were performed at least three times. For IF/IHC/HE stains, the number of independent samples is consistent with the relevant quantitative graph, the images are representative images of at least three independent replicates of the experiments.

### Reporting summary

Further information on research design is available in the Nature Portfolio Reporting Summary linked to this article.

## Data availability

All data generated or analyzed during this study are present in the main text and supplementary data. Source data are provided with this paper.

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

## Acknowledgements

This study was funded by the Major Program of National Natural Science Foundation of China (grant no. 82196821 (to H.-B.W.)), National Natural Science Foundation of China (grant no. 82271172 (to H.-B.W.), grant no. 82171150 (to D.-G.Z.) and grant no. 82002987 (to W.-Q.Y.)), Major Fundamental Research Program of the Natural Science Foundation of Shandong Province, China (grant no. ZR2021ZD40 (to H.-B.W.)), Taishan Scholar Foundation of Shandong Province (grant no. ts20130913 (to H.-B.W.)), and Natural Science Foundation of Shandong Province (grant no. ZR2020MH179 (to D.-G.Z.) and grant no. ZR2021MH385 (to N.L.)).

## Author contributions

H.-B.W., D.-G.Z., N.L., and W.-Q.Y. designed the experiments, reviewed data, and supervised the research and overall study. D.-G.Z., N.L., W.-Q.Y., and J.-H.L., collected data, interpreted results, and wrote the manuscript. W.-Q.Y., L.-G.K, and N.Z. performed in vitro experiments. J.-H.L., Y.-D.S. and X.-F.L. performed in vivo experiments. Z.-M.F., and Y.-F.L. provided critical suggestions and discussions throughout the study. All authors discussed the results and participated in manuscript preparation and editing.

## Competing interests

The authors declare no competing interests.
