## [Peer Review File · Nature Communications]

Serum/glucocorticoid-inducible kinase 1 deficiency induces NLRP3 inflammasome activation and autoinflammation of macrophages in a murine endolymphatic hydrops modelREVIEWER COMMENTS

Reviewer #1 (Remarks to the Author):

This is an interesting and well-conducted study on NLRP3 inflammasome regulation in macrophages. However, the evidence to support its role in Meniere disease (MD) is limited. The authors show evidence to support that SGK1 regulates NLRP3 inflammasome signaling and reduces macrophage inflammation, contributing to LPS-induced endolymphatic hydrops (EH).

However, MD is not equivalent to endolymphatic hydrops, since several factors can induce EH, such as sudden or progressive sensorineural hearing loss after several years. However, we agree that the modulation of SGK1/NLRP3 signaling could have some therapeutic potential for treating MD in patients with autoinflammation, which represent around 20% of MD.

This paper deserves its publication since the authors provide an extensive set of sound experiments combining animal and cell culture model to assess the regulation of NLRP3 inflammasome in macrophages.

I have some comments and questions to clarify the results and I wish that they could add more value to the study.

Title, I suggest changing the title. I do not like a broken title. Please, consider this suggestion:

SGK1 deficiency induces NLRP3 inflammasome activation and autoinflammation of macrophages in Meniere disease

Abstract

The sentence "Ménière's disease (MD) is possibly an immune-mediated disease in which inner ear macrophages may be involved" is a bit speculative. Although many studies have found that the immune response is altered in some patients with MD, it is not clear if this is a primary disorder or the abnormal response to another trigger.

The sentence "Blocking NLRP3 signaling ameliorated MD symptoms in sgk1 knockout mice" is also inaccurate, since MD symptoms in humans are episodes of vertigo associated with sensorineural hearing loss. The sgk1 KO mouse show a different phenotype with progressive hearing loss and vestibular hypofunction. This should be re-written.

However, I am in agreement with the sentence "SGK1 mediates NLRP3 inflammasome signaling, and it may participate in the MD pathogenesis", to avoid overstating the results.

Results

Figure 1b. It would be very interesting if the authors could show a lower magnification to see both structures, ampulla and macula, better.

Figure 1d is not clear. The legend indicates AN and MD, AN being used as controls? In the text, these controls are reported as healthy volunteers, and acoustic neuroma patients are not "healthy" and they should have hearing loss. Please, revise figure legend.

In Fig 1d shows that all patients with MD (N=20) had high levels of serum IL1B. This is very unusual since only 20% will be expected to have high levels of IL1B, according to Frejo et al (2018). Could you detail how patients were selected? Were the blood samples obtained during the surgery or during an episode of vertigo?

There is no clinical information about the selected patients with MD. There are 5 clinical subtypes in MD and the levels of IL1B are different according to the subgroup. A Table with the basic clinical information about the patients should be included (age, sex, age of onset,

hearing thresholds, clinical subtype, uni/bilateral, comorbidities).

Fig 1h. I suggest changing either the pink or the Red colors for better distinction.

On page 3, the authors state that they exposed the basilar membrane of the mouse cochlea to IL-1B... Why the Basilar membrane? Is there any evidence that in humans there are Macrophages in the basilar membrane? Where in the cochlea? Basal, mid or apical turn? Have they done the experiment at different turns of the cochlea? Have they found differences?

MD usually start involving low frequencies in the cochlea. Did you found any differences in NLRP3 in the basal vs apical turn of the cochlea.

Page 7, Could you please indicate the proportion of apoptotic and pyroptotic populations in SGK1-depleted cells? Fig.2i

Page 9, Typo mistake: SGK1 is a he AGC serine/threonine kinase family highly homologous to AKT that mediates

Discussion

Could you elaborate an hypothesis to explain why SGK1 levels were lower in the vestibular end organs of MD patients compared with acoustic neuroma?

A tentative hypothesis could be activation of NFkB via TWEAK and Fn14 receptor (TNFRSF12A). Did the authors measure NFkB in VRML cells from MD patients? This could be done either at gene expression level by qPCR or by IF at protein level.

References 1 and 71 are the same. Delete if it is appropriated according to instruction for authors.

Suggested additional references

1. Frejo L, Requena T, Okawa S, et al. Regulation of Fn14 Receptor and NF- κ B Underlies Inflammation in Meniere's Disease. *Front Immunol.* 2017;8(DEC):1739. doi:10.3389/fimmu.2017.01739
2. Frejo L, Lopez-Escamez JA. Cytokines and Inflammation in Meniere Disease. *Clin Exp Otorhinolaryngol* 2022. doi: 10.21053/ceo.2021.00920.

REVIEWER COMMENTS

Reviewer #1 (Remarks to the Author):

This is an interesting and well-conducted study on NLRP3 inflammasome regulation in macrophages. However, the evidence to support its role in Meniere disease (MD) is limited. The authors show evidence to support that SGK1 regulates NLRP3 inflammasome signaling and reduces macrophage inflammation, contributing to LPS-induced endolymphatic hydrops (EH).

However, MD is not equivalent to endolymphatic hydrops, since several factors can induce EH, such as sudden or progressive sensorineural hearing loss after several years. However, we agree that the modulation of SGK1/NLRP3 signaling could have some therapeutic potential for treating MD in patients with autoinflammation, which represent around 20% of MD.

This paper deserves its publication since the authors provide an extensive set of sound experiments combining animal and cell culture model to assess the regulation of NLRP3 inflammasome in macrophages.

I have some comments and questions to clarify the results and I wish that they could add more value to the study.

Title, I suggest changing the title. I do not like a broken title. Please, consider this suggestion:

SGK1 deficiency induces NLRP3 inflammasome activation and autoinflammation of macrophages in Meniere disease

Abstract

The sentence “Ménière’s disease (MD) is possibly an immune-mediated disease in which inner ear macrophages may be involved” is a bit speculative. Although many studies have found that the immune response is altered in some patients with MD, it is not clear if this is a primary disorder or the abnormal response to another trigger.

The sentence “Blocking NLRP3 signaling ameliorated MD symptoms in *sgk1* knockout mice” is also inaccurate, since MD symptoms in humans are episodes of vertigo associated with sensorineural hearing loss. The *sgk1* KO mouse show a different phenotype with progressive hearing loss and vestibular hypofunction. This should be re-written.

However, I am in agreement with the sentence “SGK1 mediates NLRP3 inflammasome signaling, and it may participate in the MD pathogenesis”, to avoid overstating the results.

Results

Figure 1b. It would be very interesting if the authors could show a lower magnification to see both structures, ampulla and macula, better.

Figure 1d is not clear. The legend indicates AN and MD, AN being used as controls? In the text, these controls are reported as healthy volunteers, and acoustic neuroma patients are not “healthy” and they should have hearing loss. Please, revise figure legend.

In Fig 1d shows that all patients with MD (N=20) had high levels of serum IL1B. This is very unusual since only 20% will be expected to have high levels of IL1B, according to Frejo et al (2018). Could you detail how patients were selected? Were the blood samples obtained during the surgery or during an episode of vertigo?

There is no clinical information about the selected patients with MD. There are 5 clinical subtypes in MD and the levels of IL1B are different according to the subgroup. A Table with the basic clinical information about the patients should be included (age, sex, age of onset, hearing thresholds, clinical subtype, uni/bilateral, comorbidities).

Fig 1h. I suggest changing either the pink or the Red colors for better distinction.

On page 3, the authors state that they exposed the basilar membrane of the mouse cochlea to IL-1B... Why the Basilar membrane? Is there any evidence that in humans there are Macrophages in the basilar membrane? Where in the cochlea? Basal, mid or apical turn? Have they done the experiment at different turns of the cochlea? Have they found differences? MD usually start involving low frequencies in the cochlea. Did you found any differences in NLRP3 in the basal vs apical turn of the cochlea.

Page 7, Could you please indicate the proportion of apoptotic and pyroptotic populations in SGK1-depleted cells? Fig.2i

Page 9, Typo mistake: SGK1 is a he AGC serine/threonine kinase family highly homologous to AKT that mediates

Discussion

Could you elaborate an hypothesis to explain why SGK1 levels were lower in the vestibular end organs of MD patients compared with acoustic neuroma?

A tentative hypothesis could be activation of NFkB via TWEAK and Fn14 receptor (TNFRSF12A). Did the authors measure NFkB in VRML cells from MD patients? This could be done either at gene expression level by qPCR or by IF at protein level.

References 1 and 71 are the same. Delete if it is appropriated according to instruction for authors.

Suggested additional references

1. Frejo L, Requena T, Okawa S, et al. Regulation of Fn14 Receptor and NF- κ B Underlies Inflammation in Meniere's Disease. *Front Immunol.* 2017;8(DEC):1739. doi:10.3389/fimmu.2017.01739
2. Frejo L, Lopez-Escamez JA. Cytokines and Inflammation in Meniere Disease. *Clin Exp Otorhinolaryngol* 2022. doi: 10.21053/ceo.2021.00920.

Point-by-point responses are provided below.

Reviewer comments are shown in black, our responses are shown in blue.

Reviewer #1 (Remarks to the Author):

This is an interesting and well-conducted study on NLRP3 inflammasome regulation in macrophages. However, the evidence to support its role in Meniere disease (MD) is limited. The authors show evidence to support that SGK1 regulates NLRP3 inflammasome signaling and reduces macrophage inflammation, contributing to LPS-induced endolymphatic hydrops (EH).

However, MD is not equivalent to endolymphatic hydrops, since several factors can induce EH, such as sudden or progressive sensorineural hearing loss after several years. However, we agree that the modulation of SGK1/NLRP3 signaling could have some therapeutic potential for treating MD in patients with autoinflammation, which represent around 20% of MD.

This paper deserves its publication since the authors provide an extensive set of sound experiments combining animal and cell culture model to assess the regulation of NLRP3 inflammasome in macrophages.

I have some comments and questions to clarify the results and I wish that they could add more value to the study.

We thank the reviewer for the positive comments and hope that all these concerns are now addressed.

1. Title, I suggest changing the title. I do not like a broken title. Please, consider this suggestion:

SGK1 deficiency induces NLRP3 inflammasome activation and autoinflammation of macrophages in Meniere disease

Response: Thank you for the constructive advice. However, your proposed title is a bit overreaching, since large component of our work are a model and not the bone fide disease. In the revised manuscript, we have changed the broken title into the complete title as “SGK1 deficiency-induced NLRP3 inflammasome activation and autoinflammation of macrophages are implicated in Ménière’s Disease pathogenesis”.

2. Abstract

The sentence “Ménière’s disease (MD) is possibly an immune-mediated disease in which inner ear macrophages may be involved” is a bit speculative. Although many studies have found that the immune response is altered in some patients with MD, it is

not clear if this is a primary disorder or the abnormal response to another trigger.

The sentence “Blocking NLRP3 signaling ameliorated MD symptoms in *sgk1* knockout mice” is also inaccurate, since MD symptoms in humans are episodes of vertigo associated with sensorineural hearing loss. The *sgk1* KO mouse show a different phenotype with progressive hearing loss and vestibular hypofunction. This should be re-written.

However, I am in agreement with the sentence “SGK1 mediates NLRP3 inflammasome signaling, and it may participate in the MD pathogenesis”, to avoid overstating the results.

Response: We have revised the text accordingly. The first sentence of the abstract reads as follows:

“Ménière's disease (MD), a multifactorial disorder of inner ear, is characterized by severe vertigo episodes and hearing loss. Although the role of immune responses in MD has been proposed, ...”.

The sentence “Blocking NLRP3 signaling ameliorated MD symptoms in *sgk1* knockout mice” was re-written to “..., which was ameliorated by blocking NLRP3 in *sgk1*-knockout mice.”

In addition, we have revised all “MD symptoms” with “EH and audiovestibular symptoms” in the revised manuscript.

3. Results

Figure 1b. It would be very interesting if the authors could show a lower magnification to see both structures, ampulla and macula, better.

Response: Thank you for pointing this out. The ampulla and macula were sampled in the labyrinthectomy surgery, during which the surgeon used a drill for cutting and an aspirator for suction and traction. To avoid damage to peripheral blood vessels and nerves, the vestibular tissue was inevitably mechanically damaged during the sampling process. The lower-magnification histochemical pictures of the ampulla and macula cannot show both structures well. However, this did not affect our detection of the expression of related molecules. In the revised manuscript, we have replaced Figure 1b with a better-structured histochemical typical image and attached its low-magnification image in the supplementary material (Supplementary Fig. 1a). Corresponding statistical data of this sample were also added to Fig. 1b. These results are shown below for your convenience:

Figure 1

Revised Fig. 1b Representative images of immunohistochemical staining and densitometric quantification of IL-1 β , IL-18, and Casp-1 in the ampulla and macula of AN (n=4) and MD patients (n=4), scale bar = 100 μ m.

Supplementary Fig.1

Revised Supplementary Fig. 1a Representative images of immunohistochemical staining of IL-1 β , IL-18, and Casp-1 in the ampulla and macula of AN (n=4) and MD patients (n=4), scale bar = 200 μ m.

4. Figure 1d is not clear. The legend indicates AN and MD, AN being used as controls? In the text, these controls are reported as healthy volunteers, and acoustic neuroma patients are not “healthy” and they should have hearing loss. Please, revise figure legend.

Response: We apologize for the confusion. We have corrected the legend of Figure 1d.

5. In Fig 1d shows that all patients with MD (N=20) had high levels of serum IL1B. This is very unusual since only 20% will be expected to have high levels of IL1B, according to Frejo et al (2018). Could you detail how patients were selected? Were the blood samples obtained during the surgery or during an episode of vertigo?

Response: The 20 patients enrolled in our study had been diagnosed with MD according to the 2015 Diagnostic criteria for Menière's disease¹. Moreover, these patients were all hospitalized and their vertigo was not well controlled; hence, they experienced frequent episodes; their peripheral blood was sampled during the active phase of MD. In the revised manuscript, we have added details regarding patient selection and specimen collection on Page 18. As shown in the revised Supplementary Table 3, most of the selected patients (approximately 70%) had a worsened functional level, mostly levels 4–6. In view of the above, these MD patients had higher levels of serum IL-1 β .

6. There is no clinical information about the selected patients with MD. There are 5 clinical subtypes in MD and the levels of IL1B are different according to the subgroup. A Table with the basic clinical information about the patients should be included (age, sex, age of onset, hearing thresholds, clinical subtype, uni/bilateral, comorbidities).

Response: We have compiled the clinical information of the MD patients, including age, sex, age of onset, hearing thresholds, clinical subtype, uni/bilateral, comorbidities and other information, in Supplementary Table 1.

Variables	MD (n=20)
Age, Mean (SD)	60.9 (10.7)
Gender (%women)	11 (55%)
Age of onset (SD)	51.3 (12.9)
Time course (years), mean (SD)	9.6(16.7)
Side, n (%)	
Unilateral	17 (85%)
bilateral	3 (15%)
Clinical subtype, n (%)	
UMD 1, classical MD	11 (55%)
UMD 2, delayed MD	1 (5%)
UMD 4, migraine	3 (15%)
UMD 5, autoimmune disease	2 (10%)
BMD 1, metachronic sensorineural hearing loss	1 (5%)
BMD 2, synchronic sensorineural hearing loss	1 (5%)
BMD 4, migraine	1 (5%)
Hearing loss at diagnosis, mean (SD)	72.5 (33.0)
Hearing stage, n (%)	
1	1 (5%)
2	2 (10%)
3	9 (45%)
4	8 (40%)
Tumarkin crisis, n (%)	2 (10%)
High blood pressure, n (%)	5 (25%)
Type 2 diabetes, n (%)	4 (20%)
Autoimmune disease, n (%)	2 (10%)
Migraine, n (%)	3 (15%)
Functional level, n%	
1	0 (0%)
2	1 (5%)
3	5 (25%)
4	2 (10%)
5	8 (40%)
6	4 (20%)

Revised Supplementary Table 1: Clinical features of patients with MD.

7. Fig 1h. I suggest changing either the pink or the Red colors for better distinction.

Response: In the revised Figure 1h, we have changed the pink colors to gray for better clarity. The revised Figure 1h is shown below for convenience:

Figure 1

Revised Fig. 1h Representative confocal microscopy images showing IL-1 β (green), SGK1 (red), IBA1 (gray), and DAPI (blue) staining in the ampulla and macula of AN and MD patients, scale bar = 10 μ m.

8. On page 3, the authors state that they exposed the basilar membrane of the mouse cochlea to IL-1B... Why the Basilar membrane? Is there any evidence that in humans there are Macrophages in the basilar membrane? Where in the cochlea? Basal, mid or apical turn? Have they done the experiment at different turns of the cochlea? Have they found differences?

MD usually start involving low frequencies in the cochlea. Did you found any differences in NLRP3 in the basal vs apical turn of the cochlea.

Response: As the level of IL-1 β , the major effector molecule of the NLRP3 inflammasome, was significantly elevated in the VEO tissue from patients with MD, we investigated whether IL-1 β produced by macrophages in the VEO affected the inner ear hair cells and vestibular neurons. The cochlear basilar membrane was selected to observe the damage to inner ear hair cells caused by IL-1 β for three reasons: First, hearing loss is frequently caused by defects in hair cells in the basilar membrane², while balance depends on the normal structure and function of the vestibular system³. Second, we have confirmed the localization of macrophages in the murine cochleae, including the basilar membrane (Supplementary Fig. 1f). Third, the basilar membrane organ culture is a well-recognized method for evaluation of inner ear injury *in vitro*. Therefore, we exposed the basilar membrane of the mouse cochlea to IL-1 β *in vitro*.

All human tissue samples in this study were obtained from patients undergoing labyrinthectomy. Usually, labyrinthectomy does not include cochlea resection; hence, we could not detect macrophages in the human basilar membrane. Alternatively, we performed immunofluorescence staining to observe macrophages in murine cochleae. As shown in Supplementary Fig. 1f, CD68-positive macrophages were widely distributed in the basal, middle, and apical turn of cochlea, primarily located in the cochlear modiolus, stria vascularis, the spiral ligament, and spiral ganglion neurons, and scattered in the basilar membrane.

We exposed all turns of the cochlea to IL-1 β *in vitro*. In the original manuscript, the results of the IL-1 β -stimulated apex were shown (Fig. 1i and 1j). In the revised Supplementary Figure 1, we present the results of middle and basal turns stimulated by IL-1 β . Compared to those in the control group, cochlear hair cells (Myosin VIIa positive) exposed to IL-1 β showed disordered arrangement, cell loss, and increased apoptosis in the apical turn (Fig. 1i and 1j), while those in the middle and basal turns showed no changes (Supplementary Fig. 1g). No significant changes in the cochlear supporting cells (Sox2 positive) were detected in apical, middle, and basal turns between the control and IL-1 β groups (Supplementary Fig. 1g).

For the reasons mentioned above, we could not detect NLRP3 expression in the human basilar membrane. Hence, we constructed an LPS-induced EH mouse model and collected the cochleae. Immunofluorescence staining showed that NLRP3-positive cells are scattered in the basal, middle, and apical turns. More NLRP3-positive staining was detected in the apex than in the base (Supplementary Fig. 3a).

Supplementary Fig.1

f

CD68 DAPI

Revised Supplementary Fig. 1f Representative confocal microscopy images showing CD68 (red) and DAPI (blue) staining in murine cochlea, scale bar = 200 μ m.

Supplementary Fig.1

Revised Supplementary Fig. 1g Immunofluorescence staining for Sox2 (green, left), myosin VIIa (red, right) and TUNEL (green, right) in the basilar membrane of middle and basal turn at P3, after treatment with IL-1 β (10 ng/mL, 24 h) *in vitro*, scale bar = 50 μ m.

Supplementary Fig.3

Revised Supplementary Fig. 3a Representative confocal microscopy images showing NLRP3 (green) and DAPI (blue) staining in cochlea of the LPS-treated mice, scale bar = 200 μ m.

9. Page 7, Could you please indicate the proportion of apoptotic and pyroptotic populations in SGK1-depleted cells? Fig.2i

Response: Propidium iodide (PI) is a nuclear and chromosome counterstain. Since PI is not permeant to live cells, it is commonly used to detect pyroptotic cells⁴. Annexin V and PI double staining was performed to assess the role of SGK1 in pyroptosis. We

have indicated the proportion of apoptotic and pyroptotic populations in SGK1-depleted cells in the Results of the revised manuscript (Page 7) and Figure 2i. Apoptotic and pyroptotic populations were found to be increased in SGK1-depleted cells compared to those in controls when exposed to inflammatory threats; 10.3% of the cells exhibited pyroptotic signatures, showing Annexin V(+)/PI(+) status and 9.78% of the cells underwent apoptosis, showing Annexin V(+)/PI(-) status. Furthermore, z-VAD-FMK and Ac-YVAD-cmk blocked pyroptosis (7.7 and 9.19%, respectively) and reduced apoptosis (5.76 and 6.91%, respectively) induced by SGK1 depletion.

Revised Fig. 2i. Dot plot showing flow cytometry analysis of control and SGK1 siRNA-transfected THP-1 cells not treated or treated with LPS and ATP with z-VAD-FMK (20 μ M, 3.5 h) or Ac-YVAD-cmk (40 μ M, 3.5 h).

10. Page 9, Typo mistake: SGK1 is a he AGC serine/threonine kinase family highly homologous to AKT that mediates

Response: We apologize for this error. We have rewritten this sentence as “SGK1 belongs to the AGC serine/threonine kinase family, highly homologous to AKT that mediates...” on Page 9.

11. Discussion

Could you elaborate an hypothesis to explain why SGK1 levels were lower in the vestibular end organs of MD patients compared with acoustic neuroma?

A tentative hypothesis could be activation of NF κ B via TWEAK and Fn14 receptor (TNFRSF12A). Did the authors measure NF κ B in VRML cells from MD patients? This could be done either at gene expression level by qPCR or by IF at protein level.

Response: While it is interesting that SGK1 expression is downregulated in MD, it is not clear why the levels of SGK1 were reduced in the vestibular end organs of patients with MD compared with those in patients with AN. One study revealed that activation of the NF- κ B pathway downregulates SGK1 expression, due to inhibition of SGK1 gene transcription via binding of the p65 NF- κ B subunit to specific sites of the SGK1

promoter⁵. Furthermore, the variant rs4947296 regulates gene expression in the TWEAK/Fn14 pathway, thereby inducing translation of NF- κ B, which is in turn associated with abnormal inflammatory response in MD⁶. Therefore, a possible tentative hypothesis is that downregulation of SGK1 in MD is attributed to the activation of NF- κ B via the TWEAK/Fn14 pathway associated with the allelic variant rs4947296. Interestingly, inhibition of SGK1 has also been reported to enhance the activity of NF- κ B and increase proinflammatory cytokine production in innate immune cells⁷, suggesting a possible negative feedback loop. Further investigation will provide greater insight into the physiological cues of SGK1 downregulation and the role of SGK1 in MD.

We have stated this hypothesis in the Discussion of the revised manuscript (Page 14–15).

To confirm this hypothesis, we measured NF- κ B expression in VRML cells from MD patients. As shown in Supplementary Figure 1d and 1e, the expression of NF- κ B p65 was significantly upregulated in patients with MD compared to that in patients with AN. Revised Figures are shown below for your convenience:

Revised Supplementary Fig. 1d Representative confocal microscopy images showing NF- κ B p65 (green), IBA1 (red) and DAPI (blue) staining in the ampulla of AN and MD patients (n=5), scale bar=25 μ m.

Revised Supplementary Fig. 1e Quantitative real-time PCR analysis of NF- κ B p65 mRNA in hVRMLs of AN (n=3) and MD patients (n=3).

12. References 1 and 71 are the same. Delete if it is appropriated according to instruction for authors.

Response: We apologize for this mistake. We have deleted the duplicate reference.

13. Suggested additional references

1. Frejo L, Requena T, Okawa S, et al. Regulation of Fn14 Receptor and NF- κ B Underlies Inflammation in Meniere's Disease. *Front Immunol.* 2017;8(DEC):1739. doi:10.3389/fimmu.2017.01739

2. Frejo L, Lopez-Escamez JA. Cytokines and Inflammation in Meniere Disease. *Clin Exp Otorhinolaryngol* 2022. doi: 10.21053/ceo.2021.00920.

Response: We thank the reviewer for this helpful suggestion. The references have been added on page 15 and page 13 and are shown below for the convenience of review:

1. Frejo, L., Requena, T, Okawa S, et al. Regulation of Fn14 Receptor and NF- κ B Underlies Inflammation in Meniere's Disease. *Front Immunol.* 2017;8(DEC):1739. doi:10.3389/fimmu.2017.01739

Reference 54 on Page 15: "Furthermore, the variant rs4947296 regulates gene expression in the TWEAK/Fn14 pathway, which induces NF- κ B translation, which is in turn associated with abnormal inflammatory response in MD ⁵⁴."

2. Frejo L, Lopez-Escamez JA. Cytokines and Inflammation in Meniere Disease. *Clin Exp Otorhinolaryngol* 2022. doi: 10.21053/ceo.2021.00920.

Reference 45 on Page 13: "In some patients with MD, high levels of IL-1 β and TNF- α might be suggestive of chronic inflammatory disorder ⁴⁵."

References:

1. Lopez-Escamez, J.A. *et al.* Diagnostic criteria for Meniere's disease. *J. Vestib. Res.* **25**, 1-7 (2015).
2. Sun, S. *et al.* Hair Cell Mechanotransduction Regulates Spontaneous Activity and Spiral Ganglion Subtype Specification in the Auditory System. *Cell* **174**, 1247-1263 e1215 (2018).
3. Brosel, S., Laub, C., Averdam, A., Bender, A. & Elstner, M. Molecular aging of the mammalian vestibular system. *Ageing Res. Rev.* **26**, 72-80 (2016).
4. Wang, Y. *et al.* Chemotherapy drugs induce pyroptosis through caspase-3 cleavage of a gasdermin. *Nature* **547**, 99-103 (2017).

5. de Seigneux, S. *et al.* NF-kappaB inhibits sodium transport via down-regulation of SGK1 in renal collecting duct principal cells. *J. Biol. Chem.* **283**, 25671-25681 (2008).
6. Frejo, L. *et al.* Regulation of Fn14 Receptor and NF-kappaB Underlies Inflammation in Meniere's Disease. *Front. Immunol.* **8**, 1739 (2017).
7. Zhou, H. *et al.* Inhibition of serum- and glucocorticoid-inducible kinase 1 enhances TLR-mediated inflammation and promotes endotoxin-driven organ failure. *FASEB J.* **29**, 3737-3749 (2015).

REVIEWER COMMENTS

Reviewer #1 (Remarks to the Author):

I have carefully revised the new version of your study and the authors's response to reviewer comments. I sincerely appreciate the efforts of the authors to improve the study, that looks much better now.

I have some minor points that I wish the authors can clarify.

The new proposed title should be modified since the words "implicated in Meniere's disease pathogenesis" is also inaccurate.

I will reformulate as "SGK1 deficiency-induced NLRP3 inflammasome activation and autoinflammation of macrophages in a murine model of endolymphatic hydrops"

The reason is that the causality has been only shown in the mouse model, but not in the VRML cells of vestibular organs. In the human samples there is an association, but causality cannot be demonstrated with these sets of experiments.

The results in human VRML demonstrate an elevation of IL1B, IL-18 and Caspase-1. Next the authors show a decreased mRNA expression of SGK1 (Fig. 1e and 1F).

However, in the lines 104-106 the statement "These analyses indicated that SGK1 downregulation is associated with NLRP3 inflammasome activation in the VEO and that it may participate in the immune pathophysiology of MD. Activation of the NF- κ B pathway has been shown to downregulate SGK1 expression". According to this hypothesis NF- κ B will need to be elevated BEFORE the activation of the NLRP3 inflammasome, and there are many triggers that can activate NF- κ B, including IL1B.

This reviewer agrees that with the current experiments SKG1 inhibition could be a mechanism to maintain autoinflammation in VRML in M, however the initial trigger of IL-1B elevation could be any bacterial or fungal antigen or viral agent. This should be added to the discussion.

The authors have revised Fig 1b and included representative images, however the legend of the macula is mistaken as ampulla also.

The response to my question 5 In Fig 1d shows that all patients with MD (N=20) had high levels of serum IL1B. This is very unusual since only 20% will be expected to have high levels of IL1B, according to Frejo et al (2018). Could you detail how patients were selected? Were the blood samples obtained during the surgery or during an episode of vertigo?

Response: The 20 patients enrolled in our study had been diagnosed with MD according to the 2015 Diagnostic criteria for Menière's disease¹. Moreover, these patients were all hospitalized and their vertigo was not well controlled; hence, they experienced frequent episodes; their peripheral blood was sampled during the active phase of MD. In the revised manuscript, we have added details regarding patient selection and specimen collection on Page 18. As shown in the revised Supplementary Table 3, most of the selected patients (approximately 70%) had a worsened functional level, mostly levels 4–6. In view of the above, these MD patients had higher levels of serum IL-1 β .

Thanks for this clarification. However, this raises a new major concern. Taking blood samples during the active phase of MD and finding elevated levels of IL-1 β is not surprising. Many acute infections, or even a vertigo attack could raise IL-1 β . The authors should obtain blood samples out of the vertigo attack to obtain basal data and demonstrate that elevated

levels of IL-1 β . Previous studies measured IL-1 β in plasma in patients with MD and found around 20-25% cases with elevated levels out of the vertigo attacks. We consider that at least 2 determinations should be performed showing high levels of IL-1 β to support an autoinflammatory process, and one must be out of the attack.

Regarding my question on the basilar membrane exposure to IL-1 β , I appreciate that the authors have performed additional experiments to compare the effect of IL-1 β on the apical, middle and basal. Interestingly, IL-1 β showed disordered arrangement, cell loss, and increased apoptosis in the apical turn (Fig. 1i and 1j), while those in the middle and basal turns showed no changes (Supplementary Fig. 1g). No significant changes in the cochlear supporting cells (Sox2 positive) were detected in apical, middle, and basal turns between the control and IL-1 β groups (Supplementary Fig. 1g).

This finding supports a differential effect of IL-1 β in the apical turn in the LPS-induced EH murine model of autoinflammation and it is major result in this study.

In figure suppl 1f, the authors state: "CD68-positive macrophages were widely distributed in the basal, middle, and apical turn of cochlea, primarily located in the cochlear modiolus, stria vascularis, the spiral ligament, and spiral ganglion neurons, and scattered in the basilar membrane." However, on line 115, they mention only in the mouse cochlea including the basilar membrane. What is the strong signal CD68+ in the upper right corner in figure 1f?

The authors also show measured NF- κ B expression in VRML cells from MD patients. As shown in Supplementary Figure 1d and 1e, the expression of NF- κ B p65 was significantly upregulated in patients with MD compared to that in patients with AN

This additional experiment is also consistent with previous studies that measured IL-1 β and NF- κ B in MD.

In figure 7h, in the WB the authors should add CASP1 to confirm the effect on sgk1+/+ and sgk-/-.

Material and methods

The authors use DMSO to resuspend LPS and also in the controls. DMSO has an anti-inflammatory effect. This could interfere the immune response

Supplementary Table 1. Please revise that the total number of patients with migraine should be 4 (because in clinical sybtype there are 3 UMD 4 and one BMD).

Line 472. The authors use EDTA tube to collect blood, and they state that separate serum, but it should be plasma.

Best regards,
Jose Antonio Lopez-Escamez
University of Granada,
Spain

Reviewer #3 (Remarks to the Author):

Potentially interesting manuscript contributing to the literature showing an importance of

NLRP3-dependent inflammation in hearing loss.

The work appears to have been conducted carefully and using appropriate models.

I have some suggestions regarding analysis of NLRP3 activity and the data presented in figures 2 and 4.

- 1) In fig 2b why didnt LPS induce NLRP3 expression in the WT cells? It ought to.
- 2) In 2D the immuno appears to show large levels of NLRP3 in the nucleus?? This is unusual.
- 3) In panels 2b and f it would be appropriate to include the blot for mature IL-1b (17kda) which shows inflammasome activation.
- 4) In figure 2H, to measure inflammasome specks the authors should immuno for ASC, which will be clearer than the NLRP3 data presented.
- 5) In fig 3a some specificity controls should be included, e.g. where NLRP3 or SGK1 is knocked down, what does the labelling show?
- 6) In figure 4a-d some quantification of the phospho data would allow a more robust interpretation.

REVIEWER COMMENTS

Reviewer #1 (Remarks to the Author):

I have carefully revised the new version of your study and the authors's response to reviewer comments. I sincerely appreciate the efforts of the authors to improve the study, that looks much better now.

I have some minor points that I wish the authors can clarify.

The new proposed title should be modified since the words "implicated in Meniere's disease pathogenesis" is also inaccurate.

I will reformulate as "SGK1 deficiency-induced NLRP3 inflammasome activation and autoinflammation of macrophages in a murine model of endolymphatic hydrops"

The reason is that the causality has been only shown in the mouse model, but not in the VRML cells of vestibular organs. In the human samples there is an association, but causality cannot be demonstrated with these sets of experiments.

The results in human VRML demonstrate an elevation of IL1B, IL-18 and Caspase-1. Next the authors show a decreased mRNA expression of SGK1 (Fig. 1e and 1f).

However, in the lines 104-106 the statement "These analyses indicated that SGK1 downregulation is associated with NLRP3 inflammasome activation in the VEO and that it may participate in the immune pathophysiology of MD. Activation of the NF-κB pathway has been shown to downregulate SGK1 expression". According to this hypothesis NF-κB will need to be elevated BEFORE the activation of the NLRP3 inflammasome, and there are many triggers that can activate NF-κB, including IL1B. This reviewer agrees that with the current experiments SGK1 inhibition could be a mechanism to maintain autoinflammation in VRML in M, however the initial trigger of IL-1B elevation could be any bacterial or fungal antigen or viral agent. This should be added to the discussion.

The authors have revised Fig 1b and included representative images, however the legend of the macula is mistaken as ampulla also.

The response to my question 5 In Fig 1d shows that all patients with MD (N=20) had high levels of serum IL1B. This is very unusual since only 20% will be expected to have high levels of IL1B, according to Frejo et al (2018). Could you detail how patients were selected? Were the blood samples obtained during the surgery or during an episode of vertigo?

Response: The 20 patients enrolled in our study had been diagnosed with MD according to the 2015 Diagnostic criteria for Menière's disease¹. Moreover, these patients were all hospitalized and their vertigo was not well controlled; hence, they experienced frequent episodes; their peripheral blood was sampled during the active phase of MD. In the revised manuscript, we have added details regarding patient selection and

specimen collection on Page 18. As shown in the revised Supplementary Table 3, most of the selected patients (approximately 70%) had a worsened functional level, mostly levels 4–6. In view of the above, these MD patients had higher levels of serum IL-1 β .

Thanks for this clarification. However, this raises a new major concern. Taking blood samples during the active phase of MD and finding elevated levels of IL-1 β is not surprising. Many acute infections, or even a vertigo attack could raise IL-1 β . The authors should obtain blood samples out of the vertigo attack to obtain basal data and demonstrate that elevated levels of IL-1 β . Previous studies measured IL-1 β in plasma in patients with MD and found around 20-25% cases with elevated levels out of the vertigo attacks. We consider that at least 2 determinations should be performed showing high levels of IL-1 β to support an autoinflammatory process, and one must be out of the attack.

Regarding my question on the basilar membrane exposure to IL-1 β , I appreciate that the authors have performed additional experiments to compare the effect of IL-1 β on the apical, middle and basal. Interestingly, IL-1 β showed disordered arrangement, cell loss, and increased apoptosis in the apical turn (Fig. 1i and 1j), while those in the middle and basal turns showed no changes (Supplementary Fig. 1g). No significant changes in the cochlear supporting cells (Sox2 positive) were detected in apical, middle, and basal turns between the control and IL-1 β groups (Supplementary Fig. 1g).

This finding supports a differential effect of IL-1 β in the apical turn in the LPS-induced EH murine model of autoinflammation and it is major result in this study.

In figure suppl 1f, the authors state: "CD68-positive macrophages were widely distributed in the basal, middle, and apical turn of cochlea, primarily located in the cochlear modiolus, stria vascularis, the spiral ligament, and spiral ganglion neurons, and scattered in the basilar membrane." However, on line 115, they mention only in the mouse cochlea including the basilar membrane. What is the strong signal CD68+ in the upper right corner in figure 1f?

The authors also show measured NF- κ B expression in VRML cells from MD patients. As shown in Supplementary Figure 1d and 1e, the expression of NF- κ B p65 was significantly upregulated in patients with MD compared to that in patients with AN. This additional experiment is also consistent with previous studies that measured IL-1 β and NF- κ B in MD.

In figure 7h, in the WB the authors should add CASP1 to confirm the effect on sgk1 $^{+/+}$ and sgk $^{-/-}$.

Material and methods

The authors use DMSO to resuspend LPS and also in the controls. DMSO has an anti-inflammatory effect. This could interfere the immune response

Supplementary Table 1. Please revise that the total number of patients with migraine should be 4 (because in clinical sybtype there are 3 UMD 4 and one BMD).

Line 472. The authors use EDTA tube to collect blood, and they state that separate serum, but it should be plasma.

Best regards,
Jose Antonio Lopez-Escamez
University of Granada,
Spain

Reviewer #3 (Remarks to the Author):

Potentially interesting manuscript contributing to the literature showing an importance of NLRP3-dependent inflammation in hearing loss.

The work appears to have been conducted carefully and using appropriate models.

I have some suggestions regarding analysis of NLRP3 activity and the data presented in figures 2 and 4.

- 1) In fig 2b why didnt LPS induce NLRP3 expression in the WT cells? It ought to.
- 2) In 2D the immuno appears to show large levels of NLRP3 in the nucleus?? This is unusual.
- 3) In panels 2b and f it would be appropriate to include the blot for mature IL-1b (17kda) which shows inflammasome activation.
- 4) In figure 2H, to measure inflammasome specks the authors should immuno for ASC, which will be clearer than the NLRP3 data presented.
- 5) In fig 3a some specificity controls should be included, e.g. where NLRP3 or SGK1 is knocked down, what does the labelling show?
- 6) In figure 4a-d some quantification of the phospho data would allow a more robust interpretation.

Point-by-point responses are provided below.

Reviewer comments are shown in black; our responses are shown in blue.

Reviewer #1 (Remarks to the Author):

I have carefully revised the new version of your study and the authors's response to reviewer comments. I sincerely appreciate the efforts of the authors to improve the study, that looks much better now.

I have some minor points that I wish the authors can clarify.

We thank the reviewer for the valuable comments and suggestion. According to the comments, we have revised our manuscript, and the comments are answered one by one as follows.

1. The new proposed title should be modified since the words "implicated in Meniere's disease pathogenesis" is also inaccurate.

I will reformulate as "SGK1 deficiency-induced NLRP3 inflammasome activation and autoinflammation of macrophages in a murine model of endolymphatic hydrops"

The reason is that the causality has been only shown in the mouse model, but not in the VRML cells of vestibular organs. In the human samples there is an association, but causality cannot be demonstrated with these sets of experiments.

Response: Thank you for your constructive advice. As per your comment, we have changed the title of the revised manuscript to "SGK1 deficiency-induced NLRP3 inflammasome activation and autoinflammation of macrophages in a murine model of endolymphatic hydrops."

2. The results in human VRML demonstrate an elevation of IL1B, IL-18 and Caspase-1. Next the authors show a decreased mRNA expression of SGK1 (Fig. 1e and 1F).

However, in the lines 104-106 the statement "These analyses indicated that SGK1 downregulation is associated with NLRP3 inflammasome activation in the VEO and that it may participate in the immune pathophysiology of MD. Activation of the NF- κ B pathway has been shown to downregulate SGK1 expression".

According to this hypothesis NF- κ B will need to be elevated BEFORE the activation of the NLRP3 inflammasome, and there are many triggers that can activate NF- κ B, including IL1B.

This reviewer agrees that with the current experiments SGK1 inhibition could be a mechanism to maintain autoinflammation in VRML in M, however the initial trigger of IL-1B elevation could be any bacterial or fungal antigen or viral agent. This should be added to the discussion.

Response: Thank you for this helpful suggestion. We have discussed the issue that you have raised in the revised manuscript.

Revision (lines 379–392, page 15): Proinflammatory cytokines, such as interleukin-1 β (IL-1 β), have been implicated in a wide variety of middle and inner ear conditions, including MD¹⁻³. Consistently, we found that an increase of IL-1 β both in plasma and vestibular end organs of MD patients. Of note, our finding that IL-1 β are elevated in 75% patients with MD of active phase, but 25% patients with MD of quiescent phase, which suggest that vertigo episodes may lead to increased IL-1 β levels. Earlier studies have proposed that IL-1 β activates NF- κ B resulting in the transcriptional activation of several genes, including inflammatory mediators^{4,5}. Thus, NF- κ B may negatively regulates SGK1, leading to MD pathogenesis through positive regulatory loops with NLRP3 and IL-1 β . In addition to immune factors, viral infections and autoimmune or structural dysfunction of the inner ear have been reported as causes of MD⁶. IL-1 β expression levels are elevated during bacterial or fungal infections⁷⁻⁹, or inflammation in patients with active autoimmune diseases^{10,11}. This indicates that, in addition to this SGK1-dependent mechanism, there may be other initial trigger of IL-1 β elevation and lead to the progression of MD.

3. The authors have revised Fig 1b and included representative images, however the legend of the macula is mistaken as ampulla also.

Response: We apologize for this mistake. We have accordingly corrected the legend of the revised Fig. 1b.

Revised Fig. 1b Representative images of immunohistochemical staining and densitometric quantification of IL-1 β , IL-18, and Casp-1 in the ampulla and macula of AN (n=4) and MD patients (n=4), scale bar = 100 μ m.

4. The response to my question 5 In Fig 1d shows that all patients with MD (N=20) had high levels of serum IL1B. This is very unusual since only 20% will be expected to have high levels of IL1B, according to Frejo et al (2018). Could you detail how patients were selected? Were the blood samples obtained during the surgery or during an episode of vertigo?

Response: The 20 patients enrolled in our study had been diagnosed with MD according to the 2015 Diagnostic criteria for Menière's disease¹. Moreover, these patients were all hospitalized and their vertigo was not well controlled; hence, they experienced frequent episodes; their peripheral blood was sampled during the active phase of MD. In the revised manuscript, we have added details regarding patient selection and specimen collection on Page 18. As shown in the revised Supplementary Table 3, most of the selected patients (approximately 70%) had a worsened functional level, mostly levels 4–6. In view of the above, these MD patients had higher levels of serum IL-1 β .

Thanks for this clarification. However, this raises a new major concern. Taking blood samples during the active phase of MD and finding elevated levels of IL-1 β is not surprising. Many acute infections, or even a vertigo attack could raise IL-1 β . The authors should obtain blood samples out of the vertigo attack to obtain basal data and demonstrate that elevated levels of IL-1 β . Previous studies measured IL-1 β in plasma in patients with MD and found around 20-25% cases with elevated levels out of the vertigo attacks. We consider that at least 2 determinations should be performed showing high levels of IL-1 β to support an autoinflammatory process, and one must be out of the attack.

Response: We thank the reviewer for this helpful suggestion. We recruited 20 patients during the quiescent phase. Consistent with the previous study, we found that IL-1 β levels were elevated in 25% of patients during the quiescent phase (**Supplementary Fig.1b**)³. Our finding that IL-1 β are elevated in 75% patients with MD of active phase, but 25% patients with MD of quiescent phase, which suggest that vertigo episodes may lead to increased IL-1 β levels.

We have compiled the clinical information of the patients during the quiescent phase in the revised supplementary Table 1. The results of plasma IL-1 β level detection have been added to the revised supplementary Fig. 1b, and this figure has been described in line 91 and discussed in line 380 in the revised manuscript.

Supplementary Fig.1

Revised Supplementary Fig. 1b ELISA of plasma IL-1 β levels in controls (n=20), MD patients during the active phase (n=20) and quiescent phase (n=20). The grid line represents the mean + 2 standard deviation of the controls.

Variables	Active phase (n=20)	Quiescent phase (n=20)
Age, Mean (SD)	60.9 (10.7)	58.9 (10.2)
Gender (%women)	11 (55%)	13 (65%)
Age of onset (SD)	51.3 (12.9)	52.0 (11.2)
Time course (years), mean (SD)	9.6 (16.7)	6.7 (5.1)
Side, n (%)		
Unilateral	17 (85%)	19 (95%)
bilateral	3 (15%)	1 (5%)
Clinical subtype, n (%)		
UMD 1, classical MD	11 (55%)	13 (65%)
UMD 2, delayed MD	1 (5%)	0 (0%)
UMD 4, migraine	3 (15%)	5 (25%)
UMD 5, autoimmune disease	2 (10%)	1 (5%)
BMD 1, metachronic sensorineural hearing loss	1 (5%)	1 (5%)
BMD 2, synchronic sensorineural hearing loss	1 (5%)	0 (0%)
BMD 4, migraine	1 (5%)	0 (0%)
Hearing loss at diagnosis, mean (SD)	72.5 (33.0)	65.5 (28.9)
Hearing stage, n (%)		
1	1 (5%)	2 (10%)
2	2 (10%)	2 (10%)
3	9 (45%)	10 (50%)
4	8 (40%)	6 (30%)
Tumarkin crisis, n (%)	2 (10%)	2 (10%)
High blood pressure, n (%)	5 (25%)	9 (45%)
Type 2 diabetes, n (%)	4 (20%)	1 (5%)
Autoimmune disease, n (%)	2 (10%)	1 (5%)
Migraine, n (%)	4 (20%)	5 (25%)
Functional level, n%		
1	0 (0%)	0 (0%)
2	1 (5%)	3 (15%)
3	5 (25%)	6 (30%)
4	2 (10%)	7 (35%)
5	8 (40%)	4 (20%)
6	4 (20%)	0 (0%)

Revised Supplementary Table 1: Clinical features of patients with MD.

5. Regarding my question on the basilar membrane exposure to IL-1 β , I appreciate that the authors have performed additional experiments to compare the effect of IL-1 β on the apical, middle and basal. Interestingly, IL-1 β showed disordered arrangement, cell loss, and increased apoptosis in the apical turn (Fig. 1i and 1j), while those in the middle and basal turns showed no changes (Supplementary Fig. 1g). No significant changes in the cochlear supporting cells (Sox2 positive) were detected in apical, middle, and basal turns between the control and IL-1 β groups (Supplementary Fig. 1g).

This finding supports a differential effect of IL-1 β in the apical turn in the LPS-induced EH murine model of autoinflammation and it is major result in this study.

Response: We thank the reviewer for such kind comments.

6. In figure suppl 1f, the authors state: “CD68-positive macrophages were widely distributed in the basal, middle, and apical turn of cochlea, primarily located in the cochlear modiolus, stria vascularis, the spiral ligament, and spiral ganglion neurons, and scattered in the basilar membrane.” However, on line 115, they mention only in the mouse cochlea including the basilar membrane. What is the strong signal CD68+ in the upper right corner in figure 1f?

Response: We thank the reviewer for pointing out this issue. We have edited the section in the revised manuscript (Lines 115–118, page 5): “Moreover, we have confirmed that CD68-positive macrophages were widely distributed in all turns of the cochlea, primarily in the cochlear modiolus, stria vascularis, spiral ligament, spiral ganglion neurons, and basilar membrane (Supplementary Fig. 1g).”

The strong signal in the upper right corner represents the bone marrow, which was encapsulated by the osseous capsule of mouse inner ear. As previous study reported¹², CD68-positive cells in the bone marrow are quite abundant. For clarity, we have labeled those corresponding parts in revised supplementary Fig.1g and shown below for the convenience of review.

Supplementary Fig.1

Revised Supplementary Fig. 1g Representative confocal microscopy images showing CD68 (red) and DAPI (blue) staining of murine cochlea, scale bar = 200 μ m. CM, cochlear modiolus. SV, stria vascularis. SL, spiral ligament. SGN, spiral ganglion neurons. BM, basilar membrane.

- The authors also show measured NF- κ B expression in VRML cells from MD patients. As shown in Supplementary Figure 1d and 1e, the expression of NF- κ B p65 was significantly upregulated in patients with MD compared to that in patients with AN

This additional experiment is also consistent with previous studies that measured IL-1 β and NF- κ B in MD.

Response: We thank the reviewer for such favorable comments.

- In figure 7h, in the WB the authors should add CASP1 to confirm the effect on *sgk1*^{+/+} and *sgk1*^{-/-}.

Response: We thank the reviewer for pointing out this issue. Accordingly, we have performed these suggested WB experiments. The expression of Caspase-1 in *sgk1*^{+/+}, *sgk1*^{-/-}, or *sgk1*^{-/-} with MCC950 in the inner tissues was determined (**Revised Fig. 7h**). The expression of cleaved Caspase-1 in the *sgk1*^{-/-} group was significantly increased compared with that in the *sgk1*^{+/+} group, and the increasing effect of *sgk1* knockdown on cleaved Caspase-1 was, at least partially, abolished when MCC950 was added. These results are also shown below for the convenience of review.

Revised Fig. 7h Western blot analysis showing NLRP3, SGK1, pro-Casp1, Casp-1 p20, pro-IL-1 β , and cleaved IL-1 β levels in the ears (n=3).

Material and methods

9. The authors use DMSO to resuspend LPS and also in the controls. DMSO has an anti-inflammatory effect. This could interfere the immune response

Response: We thank the reviewer for pointing out this issue. In our study, LPS was resuspended with 0.9% physiological saline and DMSO was used to dissolve the SGK1 inhibitor. We used GSK650394 (MCE) as an SGK1 inhibitor to explore whether SGK1 inhibition exacerbates LPS-induced EH and audiovestibular symptoms in vivo. As GSK650394 is insoluble in water, we used DMSO as its solvent according to the manufacturer's instructions. Moreover, the control mice were injected with DMSO, so as not to affect the experimental results by solvent.

We apologize for the confusion. Accordingly, we have edited these lines for clarity.

Revision (lines 462–468, page 19): Mice were challenged with LPS (2 mg/kg, Sigma) dissolved in saline through postauricular (p.a.) injection¹³ once a day for 3 days to establish the EH mouse model. The control groups were injected p.a. with an equivalent of 0.9% physiological saline.

To evaluate the effect of SGK1 inhibition, we treated the EH mice model with SGK1 inhibitor GSK650394 (10 mM, 40 μ L/d/mouse, p.a., MCE)¹⁵. According to the instructions, GSK650394 was dissolved in dimethyl sulfoxide (DMSO, Sigma) and the corresponding control group mice were treated with an equal amount of DMSO.

10. Supplementary Table 1. Please revise that the total number of patients with migraine should be 4 (because in clinical sybtype there are 3 UMD 4 and one BMD).

Response: We apologize for this mistake. We have corrected the total number of patients with migraine to “4 (20%)” in the revised supplementary Table 1. Based on the suggestion in Q4, the clinical information of patients during quiescent phase also been compiled in revised supplementary Table 1.

11. Line 472. The authors use EDTA tube to collect blood, and they state that separate serum, but it should be plasma.

Response: We apologize for this mistake. We have accordingly changed “serum” to “plasma” in the revised manuscript.

Reviewer #3 (Remarks to the Author):

Potentially interesting manuscript contributing to the literature showing an importance of NLRP3-dependent inflammation in hearing loss.

The work appears to have been conducted carefully and using appropriate models.

I have some suggestions regarding analysis of NLRP3 activity and the data presented in figures 2 and 4.

We thank the reviewer for the kind comments and helpful suggestions. According to those suggestions, we have revised our manuscript and the comments are answered one by one as follows.

1. In fig 2b why didnt LPS induce NLRP3 expression in the WT cells? It ought to.

Response: We thank the reviewer for pointing out this issue. First of all, we apologize for the misleading WB results. *Sgk1*^{+/+} mVRMLs were found to express NLRP3 in the absence of stimulation or in the presence of LPS. As NLRP3 expression was greatly increased after SGK1 knockdown, to better show the fold-change in NLRP3, we selected a more appropriate time point for NLRP3 exposure in *sgk1*^{-/-} mice, as a result, NLRP3 expression in *sgk1*^{+/+} mice appeared to be almost nonexistent. We readjusted the exposure time to show the mVRMLs of *sgk1*^{+/+} and *sgk1*^{-/-} as completely as possible. The results are shown in the revised Figure 2b.

Figure 2

Revised Fig. 2b Primary mVRML cells from *sgk1*^{+/+} or *sgk1*^{-/-} mice upon stimulation with LPS (200 ng/mL, 3 h) alone or with ATP (5 mM, 30 min). Cell lysates were subjected to western blotting.

2. In 2D the immuno appears to show large levels of NLRP3 in the nucleus?? This is unusual.

Response: We thank the reviewer for pointing out this issue. Generally, activation of macrophage NLRP3 pyrosomes occurs in the cytoplasm. Therefore, we were also surprised to find that NLRP3 was localized in both the nucleus and cytoplasm in mVRMLs, which was a very interesting finding. We have reviewed relevant literature on NLRP3 localization in cells. In THP-1 cells, studies have reported that NLRP3 is located in both the cytoplasm and nucleus (**Example 1**)¹⁴. Lin, X. et al. investigated the expression of NLRP3 in the microglia of middle cerebral artery occlusion/reperfusion (MCAO/R) and found that NLRP3 was located not only in the cytoplasm but also in the nucleus (**Example 2**)¹⁵. Moreover, Alfonso-Loeches, S. et al. reported that stimulation of astrocytes with ATP and LPS led to the recruitment of NLRP3 within the mitochondria. Strikingly, from the published figures, NLRP3 was also expressed in the nucleus (**Example 3**)¹⁶. Bruchard, M. et al. observed that in CD4⁺ TH1 cells, NLRP3 was located in the cytoplasm; in contrast, in CD4⁺ TH2 cells, NLRP3 was located in the nucleus (**Example 4**)¹⁷. This unusual nuclear localization of NLRP3 has also been observed for other NLRs, such as CIITA, which can be located in the nucleus of COS-7 cells (**Example 5**)¹⁸. These findings suggest a difference in NLRP3 localization between cell lines according to tissue heterogeneity.

In our study, NLRP3 was found in both the nucleus and cytoplasm of VRMLs of *sgk1*^{+/+} and *sgk1*^{-/-} mice (**Revised Fig. 2d, Fig. 3c**). In the process of Fig. 2d image selection, we selected a single level confocal image, which led to the NLRP3's positioning seems to be all located in the nucleus. We superimposed the original confocal scanning images of each layer (**Revised Fig. 2d**). Furthermore, we extracted the nuclear and cytoplasmic proteins of mVRMLs and found that NLRP3 was localized not only in the nucleus but also in the cytoplasm (**Revised Supplementary Fig.2c**). We speculate a difference in NLRP3 localization across cell lines according to the tissue heterogeneity, which may indicate that in mVRMLs, the cytoplasmic localization of NLRP3 may promote the assembly of inflammatory bodies, whereas its nuclear localization may be beneficial to the transcription function of inflammatory bodies. These details (lines 147–148, page 6) and discussion (lines 393–399, page 16) have been added in the revised manuscript.

Example 1
Fig. 5

Example 1. Dubey, S. et al. *Front Immunol* **9**, 195 (2018)

Example 2
Fig. 3c

Example 2. Lin, X. et al. *J Inflamm Res* 14, 2061–2078 (2021)

Example 3
Fig. 6

Example 3. Alfonso-Loeches, S. et al. *Front Cell Neurosci* 8, 216 (2014)

Example 4
Fig. 4c

Fig. 4e

Example 4. Bruchard, M. et al. *Nat Immunol* 16, 859–870 (2015)

Example 5
Fig. 4c

Example 5. Cressman, D.E. et al. *J Immunol* **167**, 3626–3634 (2001).

Figure 2

d

Revised Fig. 2d Confocal microscopy of NLRP3 (green) and DAPI (blue) staining, scale bar = 50 μ m.

Figure 3

c

Fig. 3c Immunofluorescence staining for DAPI (blue), NLRP3 (green), CD68 (red), and SGK1 (magenta) in primary mVRML cells of WT mice, which were stimulated with LPS (200 ng/mL, 3 h) and ATP (5 mM, 30 min).

Supplementary Fig.2

Revised Supplementary Fig. 2c Western blotting analysis was performed in primary mVRMLs nucleus and cytoplasm.

- In panels 2b and f it would be appropriate to include the blot for mature IL-1b (17kda) which shows inflammasome activation.

Response: We thank the reviewer for this helpful suggestion. The blot for mature IL-1b (17 kDa) have been added in revised Fig. 2b and 2f. The protein levels of proIL-1 β and cleaved IL-1 β were increased in *sgk1*^{-/-} mVRML cells and *SGKI*-knockdown THP-1 cells than in the corresponding cells of the control. These results are also shown below for the convenience of review.

Figure 2

Revised Fig. 2b Primary mVRML cells from *sgk1*^{+/+} or *sgk1*^{-/-} mice upon stimulation with LPS (200 ng/mL, 3 h) alone or with ATP (5 mM, 30 min). Cell lysates were subjected to western blotting analysis.

Figure 2

Revised Fig. 2f Control and SGK1 siRNA-transfected THP-1 cells were stimulated with LPS (200 ng/mL, 3 h) alone or with ATP (5 mM, 30 min). Cell lysates were subjected to western blotting analysis.

4. In figure 2H, to measure inflammasome specks the authors should immuno for ASC, which will be clearer than the NLRP3 data presented.

Response: We thank the reviewer for this helpful suggestion. Accordingly, we have further examined the ASC specks using immunofluorescence analysis and found an increase in ASC assembly into specks in *SGK1*-depleted THP-1 cells stimulated with LPS and ATP (**Revised Fig. 2h**). These results are also shown below for the convenience of review.

Fig.2

h

Revised Fig. 2h Representative confocal microscopy images showing ASC (green) and DAPI (blue) in Control and SGK1 siRNA-transfected THP-1 cells that stimulated with LPS (200 ng/mL, 3 h) and ATP (5 mM, 30 min), scale bar=50 μ m.

5. In fig 3a some specificity controls should be included, e.g. where NLRP3 or SGK1 is knocked down, what does the labelling show?

Response: We thank the reviewer for pointing out this issue. In Fig. 3a, neither NLRP3 nor SGK1 was knocked down. Using immunofluorescence analysis, we examined the subcellular co-localization of SGK1 with NLRP3 and found that SGK1 was co-localized with NLRP3 in LPS-primed THP-1 cells. Previous Fig. 3a showed a single-layer confocal image, in which NLRP3 expression was weak, making it appear knocked down.

To better illustrate the co-localization of SGK1 and NLRP3, the original confocal scan images of each layer were overlaid (**Revised Fig. 3a**). These results are also shown below for the convenience of review.

Figure 3

Revised Fig. 3a Immunofluorescence staining for DAPI (blue), NLRP3 (green), and SGK1 (red) in THP-1 cells that were stimulated with LPS (200 ng/mL, 3 h) and ATP (5 mM, 30 min), scale bar = 50 μ m.

6. In figure 4a-d some quantification of the phospho data would allow a more robust interpretation.

Response: We thank the reviewer for this helpful suggestion. We repeated all western blotting experiments at least three times and quantified the phosphor-Ser data. A statistical chart of the quantitative results was added to revised Fig. 4a-d and shown below for the convenience of review.

Figure 4

Revised Fig. 4a–d a Co-IP data showing the serine phosphorylation of NLRP3 from the total lysates of primary mVRML cells of *sgk1*^{+/+} or *sgk1*^{-/-} mice upon stimulation with LPS (200 ng/mL, 3 h) and ATP (5 mM, 30 min). **b** Co-IP data showing the serine phosphorylation of NLRP3 from the total lysates of control and SGK1 siRNA-transfected THP-1 cells, which were stimulated with LPS (200 ng/mL, 3 h) and ATP (5 mM, 30 min). **c** *In vitro* kinase assay results show the serine phosphorylation of NLRP3 after incubation with active SGK1-S422D or inactive SGK1-S422A via western blotting analysis. **d** *In vitro* kinase assay results show the serine phosphorylation of WT NLRP3 or NLRP3-S5A after incubation with active SGK1-S422D via western blotting analysis. **a, b** NLRP3 served as a loading control for western blotting. **c, d** GST served as a loading control for western blotting. Protein expression was quantified by gray scanning. Results are presented as mean ± SEM. Statistical analyses were carried out via two-sided t-test for a-d. * *P*<0.05, ** *P*<0.01, *** *P*<0.001. mVRMLs, mouse vestibular-resident macrophage-like cells.

References

1. Trune, D.R., Kempton, B., Hausman, F.A., Larrain, B.E. & MacArthur, C.J. Correlative mRNA and protein expression of middle and inner ear inflammatory cytokines during mouse acute otitis media. *Hear Res* **326**, 49-58 (2015).
2. Huang, C. *et al.* Up-Regulated Expression of Interferon-Gamma, Interleukin-6 and Tumor Necrosis Factor-Alpha in the Endolymphatic Sac of Meniere's Disease Suggesting the Local Inflammatory Response Underlies the Mechanism of This Disease. *Front Neurol* **13**, 781031 (2022).
3. Frejo, L. *et al.* Proinflammatory cytokines and response to molds in mononuclear cells of patients with Meniere disease. *Sci Rep* **8**, 5974 (2018).
4. Collins, T. *et al.* Transcriptional regulation of endothelial cell adhesion molecules: NF-kappa B and cytokine-inducible enhancers. *FASEB J* **9**, 899-909 (1995).
5. El Kasmi, K.C. *et al.* Macrophage-derived IL-1beta/NF-kappaB signaling mediates parenteral nutrition-associated cholestasis. *Nat Commun* **9**, 1393 (2018).
6. Rizk, H.G. *et al.* Pathogenesis and Etiology of Meniere Disease: A Scoping Review of a Century of Evidence. *JAMA Otolaryngol Head Neck Surg* **148**, 360-368 (2022).
7. Zielinski, C.E. *et al.* Pathogen-induced human TH17 cells produce IFN-gamma or IL-10 and are regulated by IL-1beta. *Nature* **484**, 514-518 (2012).
8. Hu, L., Bray, M.D., Osorio, M. & Kopecko, D.J. *Campylobacter jejuni* induces maturation and cytokine production in human dendritic cells. *Infect Immun* **74**, 2697-2705 (2006).
9. Kwak, D.J. *et al.* Intracellular and extracellular cytokine production by human mixed mononuclear cells in response to group B streptococci. *Infect Immun* **68**, 320-327 (2000).
10. Hessian, P.A., Highton, J., Kean, A., Sun, C.K. & Chin, M. Cytokine profile of the rheumatoid nodule suggests that it is a Th1 granuloma. *Arthritis Rheum* **48**, 334-338 (2003).
11. Yilmaz, M., Kendirli, S.G., Altintas, D., Bingol, G. & Antmen, B. Cytokine levels in serum of patients with juvenile rheumatoid arthritis. *Clin Rheumatol* **20**, 30-35 (2001).

12. Molitor, D.C.A. *et al.* Macrophage frequency in the bone marrow correlates with morphologic subtype of myeloproliferative neoplasm. *Ann Hematol* **100**, 97-104 (2021).
13. Li, J. *et al.* Postauricular hypodermic injection to treat inner ear disorders: experimental feasibility study using magnetic resonance imaging and pharmacokinetic comparison. *J Laryngol Otol* **127**, 239-245 (2013).
14. Dubey, S. *et al.* Withaferin A Associated Differential Regulation of Inflammatory Cytokines. *Front Immunol* **9**, 195 (2018).
15. Lin, X., Zhan, J., Jiang, J. & Ren, Y. Upregulation of Neuronal Cyindromatosis Expression is Essential for Electroacupuncture-Mediated Alleviation of Neuroinflammatory Injury by Regulating Microglial Polarization in Rats Subjected to Focal Cerebral Ischemia/Reperfusion. *J Inflamm Res* **14**, 2061-2078 (2021).
16. Alfonso-Loeches, S., Urena-Peralta, J.R., Morillo-Bargues, M.J., Oliver-De La Cruz, J. & Guerri, C. Role of mitochondria ROS generation in ethanol-induced NLRP3 inflammasome activation and cell death in astroglial cells. *Front Cell Neurosci* **8**, 216 (2014).
17. Bruchard, M. *et al.* The receptor NLRP3 is a transcriptional regulator of TH2 differentiation. *Nat Immunol* **16**, 859-870 (2015).
18. Cressman, D.E., O'Connor, W.J., Greer, S.F., Zhu, X.S. & Ting, J.P. Mechanisms of nuclear import and export that control the subcellular localization of class II transactivator. *J Immunol* **167**, 3626-3634 (2001).

REVIEWERS' COMMENTS

Reviewer #1 (Remarks to the Author):

I have revised the revised version of the submission SGK1 deficiency-induced NLRP3 inflammasome activation and autoinflammation of macrophages in a murine model of endolymphatic hydrops. The authors have responded all my questions and performed additional experiments to complete the study.

The conclusion are supported by the experimental data and the methodology is robust. This human and animal findings will be a significant contribution to the molecular understanding of inflammation in Meniere disease.

Best regards,
Jose Antonio Lopez Escamez
University of Granada

Reviewer #3 (Remarks to the Author):

The authors have addressed the points I raised with my initial review. No further comments.

REVIEWERS' COMMENTS

Reviewer #1 (Remarks to the Author):

I have revised the revised version of the submission SGK1 deficiency-induced NLRP3 inflammasome activation and autoinflammation of macrophages in a murine model of endolymphatic hydrops. The authors have responded all my questions and performed additional experiments to complete the study.

The conclusion are supported by the experimental data and the methodology is robust. This human and animal findings will be a significant contribution to the molecular understanding of inflammation in Meniere disease.

Best regards,
Jose Antonio Lopez Escamez
University of Granada

Reviewer #3 (Remarks to the Author):

The authors have addressed the points I raised with my initial review. No further comments.

Point-by-point responses are provided below.

Reviewer comments are shown in black; our responses are shown in blue.

Reviewer #1 (Remarks to the Author):

I have revised the revised version of the submission SGK1 deficiency-induced NLRP3 inflammasome activation and autoinflammation of macrophages in a murine model of endolymphatic hydrops. The authors have responded all my questions and performed additional experiments to complete the study.

The conclusion are supported by the experimental data and the methodology is robust. This human and animal findings will be a significant contribution to the molecular understanding of inflammation in Meniere disease.

We thank the reviewer for the positive comments and for accurately summarizing our work.

Reviewer #3 (Remarks to the Author):

The authors have addressed the points I raised with my initial review. No further comments.

We thank the reviewer very much for your good suggestions and kind help.